# Improving Detection of Rare Nodes in Hierarchical Multi-Label Learning

**Isaac Xu**[†,*] Ⓘ                                    *isaac.xu@dal.ca*
*Faculty of Computer Science*
*Dalhousie University*

**Martin Gillis**[†] Ⓘ                                    *martin.gillis@dal.ca*
*Faculty of Computer Science*
*Dalhousie University*

**Ayushi Sharma**                                    *ay847821@dal.ca*
*Faculty of Computer Science*
*Dalhousie University*

**Benjamin Misiuk** Ⓘ                                    *bmisiuk@mun.ca*
*Department of Geography*
*Memorial University of Newfoundland*

**Craig J. Brown** Ⓘ                                    *craig.brown@dal.ca*
*Department of Oceanography*
*Dalhousie University*

**Thomas Trappenberg** Ⓘ                                    *tt@cs.dal.ca*
*Faculty of Computer Science*
*Dalhousie University*

**Reviewed on OpenReview:** *https://openreview.net/forum?id=hf4zEWWIvE*

[†] Co-first authors.
[*] Corresponding author.

## Abstract

In hierarchical multi-label classification, a persistent challenge is enabling model predictions to reach deeper levels of the hierarchy for more detailed or fine-grained classifications. This difficulty partly arises from the natural rarity of certain classes (or hierarchical nodes) and the hierarchical constraint that ensures child nodes are almost always less frequent than their parents. To address this, we propose a weighted loss objective for neural networks that combines node-wise imbalance weighting with focal weighting components, the latter leveraging modern quantification of ensemble uncertainties. By emphasizing rare nodes rather than rare observations (data points), and focusing on uncertain nodes for each model output distribution during training, we observe improvements in recall by up to a factor of five on benchmark datasets, along with statistically significant gains in $F_1$ score. We also show our approach aids convolutional networks on challenging tasks, as in situations with suboptimal encoders or limited data [1].

---

[1]Code available at `https://github.com/DalhousieAI/Hierarchical_Multi-Label_Imb_Focal_Weighted_Loss`.

# 1 Introduction

In observing the natural world, contemplating the "relatedness" of objects can aid in understanding their properties. Hierarchies are therefore ubiquitous in the natural sciences. As machine learning techniques expand to these academic fields, there is growing interest in training hierarchical models capable of annotating in ways useful to experts (Romero et al., 2022; Hunt & Steenstrup Pedersen, 2023; Katija et al., 2022; Blondin et al., 2024; Lowe et al., 2025; Battach et al., 2025). For this work, we describe broader higher-level categories in the hierarchy as "ancestor" or "parent" nodes, while the lower-level detailed categories stemming from these are referred to as "descendant" or "child" nodes. More generally, hierarchies can be understood as directed acyclic graphs (DAGs), and can be tree-like (where nodes only possess a singular parent) or otherwise (where they can be descended from multiple parents). Finally, there are almost always potential hierarchical associations inherent in data, even if not annotated in this manner.

Another property, the multi-label characteristic of the data, may also appear in datasets. Multi-label refers to the possibility of multiple classifications existing for a single observation. A contemporary challenge for machine learning is the need for big data and the lack of a singular large dataset in fields traditionally accustomed to working with small and unstandardized hand-annotated datasets (Kuznetsova et al., 2020; Lowe et al., 2025). As researchers work to amalgamate these potentially multi-label datasets, they may be forced to either exclude some of these datasets or, if additional processing to separate the annotations is not available, accept the amalgamated dataset as also multi-label. Therefore, it seems reasonable that more multi-label datasets will emerge as the number of amalgamated datasets grows in various academic fields.

Combining hierarchical and multi-label properties transforms the data modelling process into a hierarchical multi-label (HML) learning task. Within HML research, classification tasks may be further divided into full depth (HML-FD) and partial depth (HML-PD). Full-depth refers to datasets where each observation is annotated to the maximum depth permitted by the hierarchy. Whereas, partial-depth permits annotations terminating at higher-level categories, prior to reaching the leaf nodes.

In HML data, class imbalance and rare nodes arise naturally from the hierarchical structure, leading to long-tailed distributions. Rare nodes often represent fine-grained distinctions crucial for research. For example, in seafloor classification, one of our areas of interest, detecting rare species or habitats, can indicate important environmental changes. This valuable information may then be used to guide government policies concerning maritime economies. Similarly, in medical domains, rare gene products identified through hierarchical classification can help diagnose disease. Thus, handling imbalance and ensuring that rare nodes are meaningfully incorporated into HML models is essential for producing insights that are both scientifically relevant and actionable.

Addressing imbalance in the HML context, in order to attain deeper and more comprehensive predictions, remains a relatively under-explored area and is a difficult problem even for modern methods — with the majority of nodes in common benchmark datasets going unpredicted (Wehrmann et al., 2018; Giunchiglia & Lukasiewicz, 2020; Pereira et al., 2018; 2021). Existing works such as Pereira et al. (2018; 2021) apply a resampling approach. This preference may be a result of the relationship between HML and pure multi-label problems, and the popularity of resampling methods in multi-label imbalance literature (Tarekegn et al., 2021). Furthermore, any observation-based approach risks overemphasizing common nodes due to the conjoined nature of annotations (Pereira et al., 2021). This problem is exacerbated by the hierarchical nature, where it is typical for parent nodes to be common, while their child nodes are rare. By emphasizing the entire observation, on the basis that it contains a rare component in its annotation, additional emphasis on the already common components present becomes nearly inevitable. Therefore, an observation-based approach is suboptimal in that it must also add to a problem it is simultaneously attempting to solve. Instead, we propose looking at the problem through a node-based lens, where the emphasis on nodes is independent of the distribution of observations and depends on the node frequencies.

Finally, we look to further enhance this effect by utilizing the concept of model uncertainty in a manner inspired by focal loss (Lin et al., 2020). We propose to emphasize nodes characterized by increased uncertainty during training. To this end, we study and apply various ways of reliably quantifying uncertainty in modern literature (Kendall & Gal, 2017; Schweighofer et al., 2023; Gillis et al., 2025b).

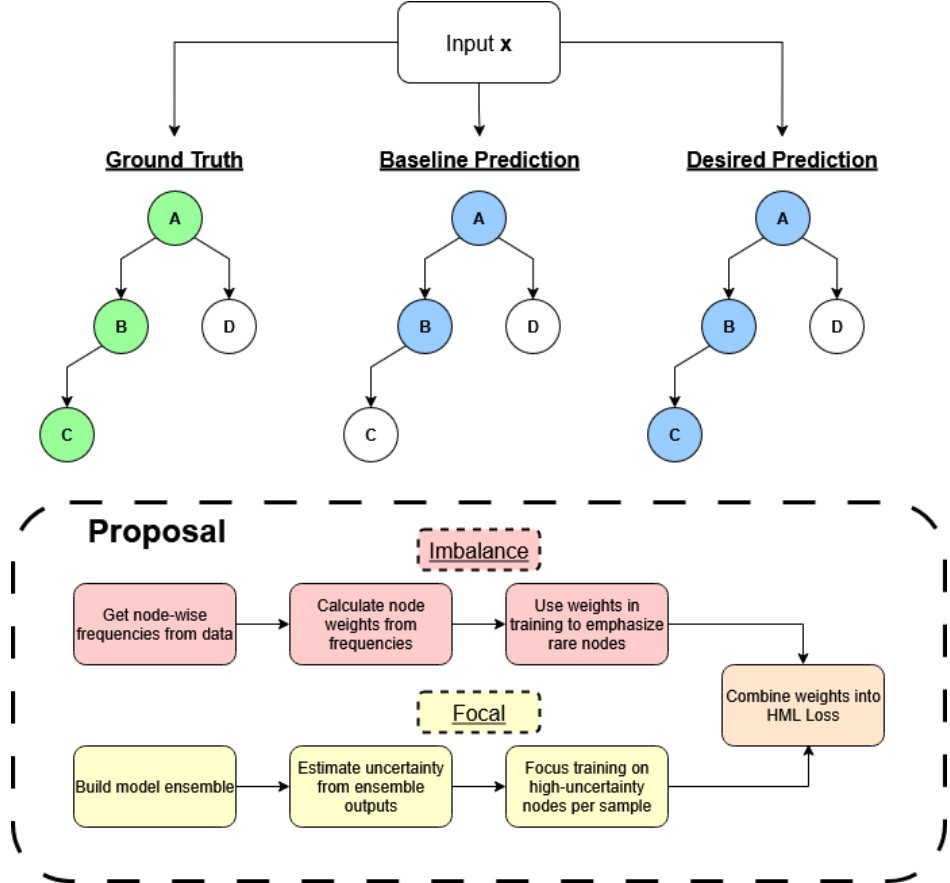

Figure 1: **Overview of HML Rare Node Problem and Proposed Weighted Approach.** The proposed approach towards detecting rare descendant nodes in HML problems consists of two branches. The imbalance branch focuses on a node-wise explorationweighting system independent of sample frequency, while the focal branch employs modern measures of uncertainty in determining challenging nodes for the model ensemble.

Our main contributions are:

1. **Imbalance weighting.** Examination of the different ways imbalance weighting may be defined and implemented to improve hierarchical classification. Evidence is provided for which techniques are effective at fulfilling the objectives in the problem statement.
2. **Focal weighting.** The concept of uncertainty, as defined in contemporary literature, is used to augment a focal term, adapted for the HML context. Evidence is provided that incorporating this uncertainty term is beneficial for improving rare node detection.

An overview of the rare node detection problem and our proposed approach is available in Figure 1.

## 2 Background

### 2.1 Learning with Hierarchical Constraint

One of the first requirements in modelling HML data is satisfying the hierarchical constraint — that is, the probability for the parent's presence in an observation must be greater than or equal to the probability of any descendants. Within the literature, there exists a number of ways to address this matter (Wehrmann

et al., 2018; Giunchiglia & Lukasiewicz, 2020). For our work, we apply Coherent Hierarchical Multi-Label Classification Neural Network (C-HMCNN) (Giunchiglia & Lukasiewicz, 2020), which remains a robust and competitive method, as a framework to demonstrate our experiments. The essence of the approach is in translating the hierarchical information into an adjacency matrix $\mathbf{A} \in \mathbb{R}^{N \times N}$, where $N$ is the number of hierarchical nodes. Each element in $\mathbf{A}$ is defined as:

$$\mathbf{A}_{ij} = \begin{cases} 1 & \text{if } j \in S_i, \\ 0 & \text{otherwise.} \end{cases} \tag{1}$$

Where $S_i$ is the set of hierarchical descendants for node $i$, and by definition, node $i$ is a descendant of itself. Each row $i$ in the matrix $\mathbf{A}$ then acts as a filter for an expanded model output, which turns off all the predictions that are outside of node $i$'s direct descendant line. Finally, a max operation taken along the rows ensures that each prediction for a node is the $\max(p(\text{node}_i), p(\text{node}_j))$, thereby satisfying the hierarchical constraint. The loss, referred to in Giunchiglia & Lukasiewicz (2020) as "max constraint loss" ($\mathcal{L}_{\text{MC}}$), involves applying this constraint mechanism in the fashion below, depicted with tensor notation for clarity. We define:

| | |
|---|---|
| $\mathbf{X} \in \mathbb{R}^{B \times F}$ | data input, $B$ = batch size, $F$ = number of features; |
| $\mathbf{Y} \in \mathbb{R}^{B \times N}$ | ground truth annotations; |
| $\theta : \mathbb{R}^{B \times F} \to \mathbb{R}^{B \times N}$ | model parameters; |
| $\mathbf{R} \in \mathbb{R}^{B \times N \times N}$ | batchwise adjacency tensor for hierarchy (batched version of $\mathbf{A}$); |
| $f_{\text{CM}} : \mathbb{R}^{B \times N} \times \mathbb{R}^{B \times N \times N} \to \mathbb{R}^{B \times N}$ | function enforcing the hierarchical constraint, takes predicted probabilities and $\mathbf{R}$, returns constrained probabilities; and |
| $\text{BCE} : \mathbb{R}^{B \times N} \times \mathbb{R}^{B \times N} \to \mathbb{R}^{B \times N}$ | unreduced binary cross-entropy, takes predictions and targets, returns a loss for each node (reduction occurs prior to backpropagation). |

$$\widetilde{\mathbf{Y}}_{\text{A}} = f_{\text{CM}}\big(\theta(\mathbf{X}), \mathbf{R}\big) \tag{2}$$

$$\widetilde{\mathbf{Y}}_{\text{B}} = f_{\text{CM}}\big(\mathbf{Y}\theta(\mathbf{X}), \mathbf{R}\big) \tag{3}$$

$$\widetilde{\mathbf{Y}} \;= (1 - \mathbf{Y})\widetilde{\mathbf{Y}}_{\text{A}} + \widetilde{\mathbf{Y}}_{\text{B}} \tag{4}$$

$$\mathcal{L}_{\text{MC}} = \text{BCE}(\widetilde{\mathbf{Y}}, \mathbf{Y}) \tag{5}$$

One particularity in this process is the asymmetrical treatment of predictions for positive nodes (nodes that are labelled as present in an observation) and predictions for negative nodes (nodes that are not present in an observation). If $\widetilde{\mathbf{Y}}_{\text{A}}$ is interpreted as the predicted component for negative nodes, we see that it is filtered in Equation 4, whereas the predicted component for positive nodes, $\widetilde{\mathbf{Y}}_{\text{B}}$, is filtered in Equation 3. This difference accounts for the possibility of model predictions for negatively annotated descendant nodes that happen to have positively annotated parents. In this case, it would be undesirable for the positive parent to have its prediction influenced by that of the negative child, which is why it is filtered prior to applying the constraint function in Equation 3. The recombined predicted probabilities (Equation 4) are then applied to calculate binary cross-entropy (BCE) loss (Equation 5). In Giunchiglia & Lukasiewicz (2020), an explicit example is presented for attempting HML directly with BCE. Without this constraint process, it is shown that gradients can point in an incorrect direction for model predictions which violate hierarchy. One last point to note is that $\widetilde{\mathbf{Y}}$ is used only for training. For inference, $\widetilde{\mathbf{Y}}_{\text{A}}$ from Equation 2 is used directly.

## 3 Methods

Our proposal to better detect rare nodes in HML observations involves augmenting the hierarchical loss with two components: a class imbalance weighting and a focal weighting, inspired by focal loss (Lin et al., 2020). We define:

$f \in \mathbb{R}^N$ — vector of occurrence frequencies for each node within a dataset;

$\widetilde{\mathbf{W}}(\mathbf{Y}, f) \in \mathbb{R}^{B \times N}$ — set of weights accounting for HML imbalance — each node has its own unique value, repeated across the batch;

$\widetilde{\mathbf{W}}_0 \in \mathbb{R}^{B \times N}$ — minimum allowed weight (information gate) for imbalance weighting — a repeated value added to each element in the imbalance weights;

$\mathbf{U}(\Theta(\mathbf{X})) \in \mathbb{R}^{B \times N}$ — set of weights attained from ensemble of models ($\Theta$) based uncertainties, for each sample and node;

$\mathbf{U}_0 \in \mathbb{R}^{B \times N}$ — minimum allowed information for uncertainty gating — a repeated value added to each element in the uncertainty weights;

$k \in \mathbb{R}$ — scalar exponential factor to emphasize nodes with extreme uncertainties.

$$\mathcal{L}_{\text{focal}} = \underbrace{\left( \widetilde{\mathbf{W}}_0 + \widetilde{\mathbf{W}}(\mathbf{Y}, f) \right)}_{\text{imbalance weighting}} \underbrace{\left( \mathbf{U}_0 + \mathbf{U}\big(\Theta(\mathbf{X})\big)^k \right)}_{\text{focal weighting}} \mathcal{L}_{\text{MC}} \tag{6}$$

### 3.1 Imbalance Weighting

As a starting point, we apply a conventional weighting method used in standard multi-class tasks:

$$w_i = \frac{N_{\text{obs}}}{N_{\text{classes}} \cdot n_i} = (N_{\text{classes}} f_i)^{-1}, \tag{7}$$

where $n_i$ denotes the total occurrences of a node and its descendants,[2] $N_{\text{obs}}$ is the number of data points, and $N_{\text{classes}}$ is the number of classes. There is some flexibility in defining $N_{\text{classes}}$ under a binary representation in HML: it could be defined either as two, corresponding to the presence or absence of a node for a given observation, or as the total number of nodes in the hierarchy. Empirically, we find that setting $N_{\text{classes}}$ equal to the number of hierarchical nodes leads to improved model performance. However, both definitions systematically assign extremely small weights to common nodes while assigning large weights to rare nodes, which can impede effective learning. To mitigate this issue, we rescale the weights to enforce a zero lower bound and introduce a minimum weight parameter $\tilde{w}_0$, ensuring that common nodes continue to contribute informative gradients during training:

$$\tilde{w}_i = \tilde{w}_0 + w_i \frac{w_i - w_{min}}{w_{max} - w_{min}}. \tag{8}$$

Since presence and absence encode different information, node weights should depend on their positive or negative annotations in each observation. In practice, we observe that introducing imbalance weights for negative annotations does not improve performance across the formulations we evaluated. Accordingly, we define the node weights as follows:

$$\tilde{w}_i(y_i) = \begin{cases} \tilde{w}_i & \text{if } y_i = 1, \\ 1 & \text{otherwise.} \end{cases} \tag{9}$$

Hence, in the abbreviated tensor notation in Equation 6, the imbalance component contains the term $\widetilde{\mathbf{W}}$ as a function of $\mathbf{Y}$ and $f$.

---

[2]Throughout, $n_i$ refers to the aggregate count for a node and all its descendants.

### 3.2 Focal Weighting

Focal loss was originally developed for dense object detection or image segmentation, where annotations are often strongly imbalanced between items or pixels of interest and background. To address these conditions, Lin et al. (2020) introduced a loss term $(1 - p(y|x;\theta))^k$, which determines the amount of weight given to an annotation based on the model's confidence — interpreted as $p(y|x;\theta)$. Over time, it is expected that the model's growing confidence would prompt the training process to ignore the common background and instead focus on the rarer annotations. The degree to which extremes are emphasized or ignored is modulated by the exponential parameter $k$. As in these tasks, HML tends to possess reliable but rare manual annotations for nodes deep within a label hierarchy, yielding a problem we believe can benefit from a focal-style weighting.

We define uncertainty as $(1 - c)$, where $c$ is confidence. Furthermore, we note that $p(y|x;\theta)$ is not necessarily a reliable measure of model confidence (Hendrycks et al., 2019; Costa et al., 2024). Therefore, we introduce adjustments to the uncertainty term, motivated by developments in uncertainty quantification literature. In total, four plausible uncertainty quantities are explored. To begin, we create an ensemble of models, $\Theta$, to capture the distribution of models, $p(\theta|\mathbf{X}, \mathbf{Y})$, for a classification task — with $\mu$ being the Bayesian Model Average (BMA) for this ensemble:

$$\mu = \hat{p}(y|x;\Theta), \quad \sigma^2 = \text{Var}[\hat{p}(y|x;\Theta)]; \tag{10}$$

$$\mu_{\max} = \max(\mu, 1 - \mu); \text{ and} \tag{11}$$

$$\hat{\mu}_{\max} = 2\left(\mu_{\max} - 0.5\right), \quad U_{\text{bBMA}} = 1 - \hat{\mu}_{\max}. \tag{12}$$

By taking the BMA, we apply a confidence and uncertainty metric which eliminates some of the individual noise stemming from independent models, thereby making the quantity more robust. An important point is the use of $\hat{\mu}_{\max}$ rather than $\mu$ directly, as smaller predicted probabilities in binary representation do not just indicate unconfident predictions in classes, but also confident negative predictions. We refer to this measure as the binary Bayesian Model Average (bBMA), with its uncertainty counterpart being the first new focal term candidate.

Our second candidate is a signal-to-noise ratio (SNR) based metric, referred to as the Gated Margin Uncertainty (GMU) (Gillis et al., 2025b), intended to capture variance across model predictions:

$$\mu_{\max^{(2)}} = \min\left(\mu, 1 - \mu\right); \tag{13}$$

$$\text{SNR} = \frac{\mu_{\max} - \mu_{\max^{(2)}}}{\sigma_{\max} + \sigma_{\max^{(2)}}}; \text{ and} \tag{14}$$

$$U_{\text{GMU}} = 1 - \left(\hat{\mu}_{\max}(1 - e^{-\text{SNR}})\right). \tag{15}$$

In Equation 13, we use $\mu_{\max^{(2)}}$, defined as the probability of the second largest predicted class, which in the binary case is also the minimum. The interpretation of the SNR is that it incorporates a notion of how much the models in $\Theta$ agree with each other. Scaling this quantity by $\hat{\mu}_{\max}$ then incorporates a notion of "confidence" to the measure. This new gated confidence term is therefore large (close to one), given high certitude and agreement between the models, and is small (close to zero), given low certitude or little agreement between the models.

More generally, in the literature, uncertainty is typically decomposed into two forms: aleatoric (data) and epistemic (model) (Kendall & Gal, 2017; Wimmer et al., 2023; Schweighofer et al., 2023). Aleatoric refers to the uncertainty inherent in the data resulting from the collection method. Consequently, aleatoric uncertainty cannot be reduced with the collection of more data. Therefore, it should, in theory, be pointless to focus on

nodes with high aleatoric uncertainty. Instead, epistemic uncertainty — the uncertainty of the true model generating the data phenomenon — should be reducible with the acquisition of more representative data or perhaps a review and focus on rare pieces of existing data.

In a recent work, Schweighofer et al. (2023), model uncertainty $U_{\text{epistemic}}$ is defined (in our terms) as:

$$U_{\text{epistemic}} = \mathbb{E}_{p(\theta|\mathbf{X})} \left[ \mathbb{E}_{p(\tilde{\theta}|\mathbf{X})} \left[ D_{\text{KL}}\big(p(y|x;\theta) \,\|\, p(y|x;\tilde{\theta})\big) \right] \right]. \tag{16}$$

This quantity can be interpreted as a matrix comparing the Kullback–Leibler (KL) divergence for the probabilistic output of every model in $\Theta$, with that of every other model in the set, with the expectation taken across the matrix dimensions:

$$U_{\text{epistemic}} = \frac{1}{N(N-1)} \sum_{\theta=1}^{N} \sum_{\tilde{\theta}=1}^{N} D_{\text{KL}}\big(p(y|x;\theta) \,\|\, p(y|x;\tilde{\theta})\big); \tag{17}$$

with $N$ referring to the number of models in $\Theta$, or ensemble size.

In addition to this formulation of epistemic uncertainty, we experiment with a variant where the KL-divergence is switched for the Jensen-Shannon divergence, as a measure bounded by one, which can perhaps add interpretability and may perform better when scaling with other quantities in the loss.

As before, we define a minimum weight in conjunction with the uncertainty term due to empirically observed benefits in training, leading to the focal component in the loss, as defined in Equation 6. Lastly, focal weighting is computed without gradients, as existing works suggest that training based on ensemble-derived quantities with gradients carries greater risk of ensemble collapse (Lee et al., 2015).

### 3.3 Metrics

Micro-averaged precision (AP) score is commonly used to evaluate HML classification performance. This metric is the rectangular approximation of the area under the precision-recall curve (AUPRC). Its popularity is inherited from multi-label research, in which thresholding model outputs can be challenging (Bi & Kwok, 2011). While it can be informative, AP tends to be overly optimistic, especially in sparsely annotated data. Furthermore, the AP score evaluates the dataset as a whole and cannot provide insightful information on the model's performance at a node level. For practical purposes, we also value assessment of the model's predictions on data (not just probabilities), which cannot be easily attained without thresholding.

As a result of these motivations, we include the node-wise $F_1$, precision, recall, and binarized average precision score (Bin. AP) as part of the model evaluation. The $F_1$, precision, and recall are calculated for each data point using the true positives, false positives, and false negatives tabulated for each node. The Bin. AP score is simply the AP score evaluated on model predictions thresholded at 0.5, taking $\geq 0.5$ as a positive prediction for a node, and $< 0.5$ as a negative prediction.

## 4 Related Works

Although our work focuses specifically on addressing HML imbalance, we consider it to be derivative of addressing multi-label imbalance more generally. Broadly speaking, approaches in this setting can be divided into four groups: resampling, classifier adaptation, ensemble approaches, and cost-sensitive algorithms (Tarekegn et al., 2021). Resampling algorithms typically look to determine observations that contain either rare or common annotations and over- or under-sample them, respectively (Charte et al., 2015a;b). Classifier adaptation approaches consist of dedicated algorithms designed to train the model to learn the imbalance prior. These approaches are varied and include examples such as Chen et al. (2006), where multi-label problems are broken down into two-class subproblems and then recombined for inference. Zhang et al. (2022) propose an ensemble approach, in which correlations between class annotations are exploited by converting multi-label binary datasets into a trinary classification task, predicting the presence of one class in conjunction

with another. The final output is then based upon the aggregation of the confidences from the binary and multi-class models. Finally, as an example of a cost-sensitive approach, Daniels & Metaxas (2017) employ Hellinger distance (Hellinger, 1909), as an objective for random forests. Among these families of approaches, resampling remains the most popular (Tarekegn et al., 2021).

Reflecting the more general multi-label trend, the most popular existing approaches to addressing HML imbalance appear to be resampling-based. These have focused on either converting HML to a multi-label problem (Pereira et al., 2018) or implementing resampling schemes such as hierarchical random oversampling (HROS-PD) (Pereira et al., 2021). Pereira et al. (2018) convert HML to a multi-label problem by treating each node within a hierarchy as a multi-label class. A multi-label resampling approach is then applied. They compare popular multi-label imbalance methods such as LPROS/US (Charte et al., 2015a), MLROS/US (Charte et al., 2015b), multi-label SMOTE (Charte et al., 2015c), and MLeNN (Liu et al., 2022), concluding that LPROS held the strongest performance.

In Pereira et al. (2021), the team proposes an approach that determines the imbalance of the hierarchical nodes and then resamples them to an acceptable threshold. In their work, it was discovered that the oversampling approach for rare nodes (HROS-PD) outperforms the undersampling approach (HRUS-PD). However, the method is limited in two ways. First, it was designed to handle tree-like hierarchies and therefore would not perform optimally for directed acyclic graph (DAG) hierarchies (as we would expect and demonstrate on our GO datasets). Furthermore, oversampling annotations conjoined with common nodes risks further oversampling already abundant nodes.

In our work, we include the HML to multi-label/LPROS method and HROS-PD as benchmarks. Following the algorithm as described in (Pereira et al., 2021), we implement the approach from scratch and include it with our code. By disconnecting the weighting and confidence for each node from their observations, we also address the issues of working in both a tree-like or DAG-like hierarchical environment, as well as avoid overemphasizing already abundant nodes.

## 5 Experiments

The experiments section is divided into four parts: we first introduce the datasets, then analyze imbalance weighting on gene product data, followed by an evaluation of focal weighting on the same datasets, and finally compare both methods on benthic imagery data. For gene product data, we use model ensembles to quantify uncertainty. For imagery data, we apply a shared encoder with an ensemble of output heads, a regularization strategy adapted for uncertainty estimation (Lee et al., 2015; Gillis et al., 2025a). Hyperparameters are provided in Appendix A.

### 5.1 Datasets

We evaluate on 16 gene product datasets, evenly divided into two sets, with each set annotated under two distinct classification frameworks: Functional Catalogue (FUN) (Ruepp et al., 2004) and Gene Ontology (GO)[3] (Ashburner et al., 2000). These two schemes were developed to catalogue gene products and proteins, and in the case of GO, the genes themselves as well. These datasets are popular benchmarks in HML works (Bi & Kwok, 2011; Giunchiglia & Lukasiewicz, 2020; Tarekegn et al., 2021; Pereira et al., 2018; 2021).

There are two key differences between FUN and GO. First, FUN contains hierarchical structures up to a depth of six (we count root node(s) as depth zero), with datasets typically containing 500 nodes, except for Eisen (FUN), which uses 462. GO contains over 4,000 nodes, with Eisen (GO) using 3,574 nodes. Nodes for GO are spread across 14 hierarchical levels. Furthermore, while FUN is tree-like, GO is a DAG.

We also make use of a subset of BenthicNet (Lowe et al., 2025), an image dataset cataloguing benthic habitats. The images we use are of Echinodermata, following the echinoderm branch of the Collaborative and Automated Tools for Analysis of Marine Imagery (CATAMI) biota hierarchical tree (Althaus et al., 2015) — a classification scheme developed for the visual recognition of underwater biota and seafloor characteristics. The hereby dubbed "BenthicNet-E" contains 13 nodes, up to a depth of two, making it a comparatively

---

[3]An introduction to GO is available at `https://geneontology.org/`.

simpler structure than FUN or GO. The BenthicNet-E dataset contains a total of 5,077 training data points and 1,650 test data points, inherited from the original full BenthicNet partitions, which considered geospatial relations between observations (Lowe et al., 2025). The size and focus of BenthicNet-E is typical for a small locally sourced dataset in the ocean sciences (Brown et al., 2012; Zhou et al., 2023), and is therefore similar in scope to popular problems in the field.

## 5.2 Imbalance Weighting on Gene Product Data

Performances for the FUN category of datasets are presented in Table 1. Since they show similar trends, results for the GO datasets are in Appendix C.

These results suggest that the Bin. AP score, $F_1$, and especially recall scores benefit significantly from node-wise weighting, with recall improving by a factor of five over the control, on certain datasets using $\tilde{w}_0 = 0.25$. However, it appears that in a few of these datasets, the gain in recall comes with a possible trade-off in precision — although the difference is often within two standard deviations ($\sigma$). Additionally, in some cases, imbalance weighting may even improve precision (as seems to be the case for Derisi, Gasch-2, and SPO).

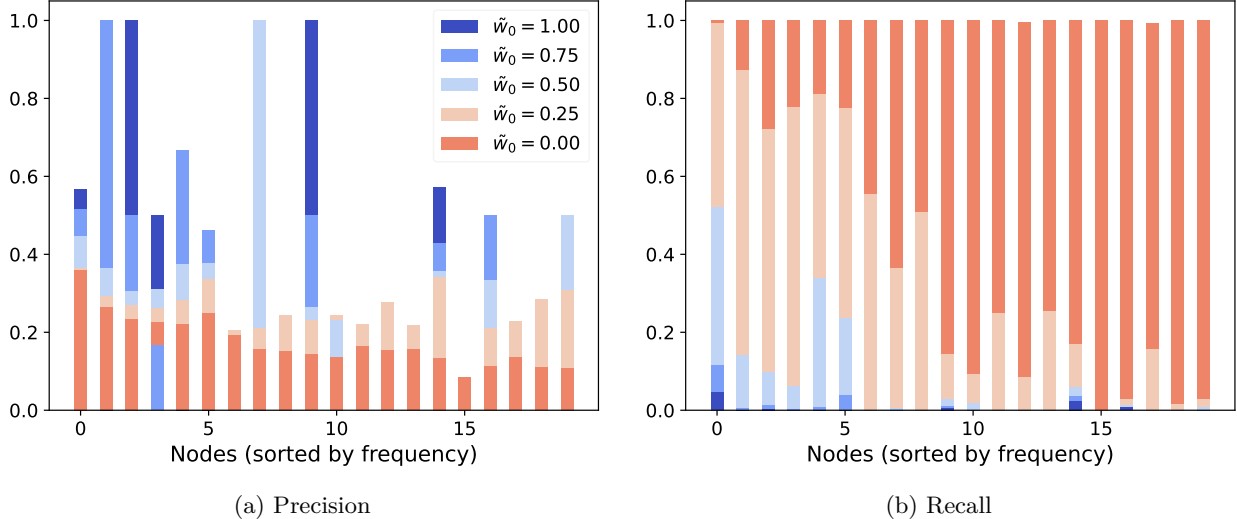

(a) Precision          (b) Recall

Figure 2: **Comparison of the Effects of Varying $\tilde{w}_0$ on Derisi (FUN).** Plotted are node-wise precision and recall scores across the top-20 most frequent nodes in their respective datasets.

In Figure 2, we note the behaviour of precision and recall on the top-20 most frequent nodes in Derisi with varying $\tilde{w}_0$. Firstly, we observe that there appears to be a trend of decreasing precision as we decrease $\tilde{w}_0$. Gains in precision appear to be due to the detection of nodes which were previously being ignored at higher $\tilde{w}_0$ values. Furthermore, we see that recall improves dramatically with almost all presented nodes, being perfectly recalled at $\tilde{w}_0 = 0$. Nonetheless, we find that if we lower $\tilde{w}_0$ to this degree, too much precision is lost, resulting in a decrease in the overall $F_1$ score. Empirically, we find $\tilde{w}_0 = 0.25$ to be a reasonable value in boosting overall $F_1$. The impact of $\tilde{w}_0$ is explored in greater resolution in Appendix H.

This observed phenomenon is perhaps counterintuitive. It may seem surprising that smaller weights ($< 1$) on positive instances for common nodes should improve recall and lower precision. Generally, we would expect that assigning a lower priority to positive annotations for a node reduces the model's tendency to predict positively, except in the most certain cases, thereby decreasing recall and increasing precision. We reason that due to the hierarchical constraint, the model may learn to classify these higher-level nodes indirectly through their descendant nodes. Subsequently, their recall increases at the cost of precision. The more we lower $\tilde{w}_0$, the more we encourage indirect node learning in this fashion.

Table 1: **HML Method Comparisons (FUN)**. Different strategies in addressing HML imbalance are compared for the FUN datasets: no method used (None), HROS-PD, and node weighting using $\tilde{w}_0$ values of 0.25 and 0.50. Means that are significantly ($> 2\sigma$) greater than all others for a dataset bolded.

| Dataset (FUN) | Method | $F_1$ | Precision | Recall | Bin. AP | AP |
|---|---|---|---|---|---|---|
| CELLCYCLE | NONE | $2.61 \pm 0.09$ | $9.81 \pm 0.70$ | $1.74 \pm 0.06$ | $6.26 \pm 0.11$ | $25.68 \pm 0.13$ |
| | LPROS | $2.98 \pm 0.18$ | $9.70 \pm 0.80$ | $2.13 \pm 0.19$ | $6.42 \pm 0.10$ | $24.81 \pm 0.25$ |
| | HROS-PD | $2.70 \pm 0.25$ | $9.57 \pm 0.90$ | $1.88 \pm 0.22$ | $6.18 \pm 0.12$ | $25.57 \pm 0.16$ |
| | $\tilde{w}_0 = 0.25$ | $\mathbf{4.85} \pm 0.11$ | $9.13 \pm 0.36$ | $\mathbf{4.59} \pm 0.11$ | $\mathbf{10.21} \pm 0.08$ | $25.45 \pm 0.10$ |
| | $\tilde{w}_0 = 0.50$ | $3.54 \pm 0.08$ | $9.34 \pm 0.36$ | $2.68 \pm 0.07$ | $8.20 \pm 0.08$ | $25.69 \pm 0.12$ |
| DERISI | NONE | $0.46 \pm 0.03$ | $1.75 \pm 0.21$ | $0.34 \pm 0.03$ | $2.67 \pm 0.08$ | $19.58 \pm 0.10$ |
| | LPROS | $0.50 \pm 0.02$ | $1.96 \pm 0.16$ | $0.35 \pm 0.02$ | $2.66 \pm 0.08$ | $19.18 \pm 0.10$ |
| | HROS-PD | $0.44 \pm 0.02$ | $1.82 \pm 0.41$ | $0.33 \pm 0.01$ | $2.70 \pm 0.09$ | $19.39 \pm 0.11$ |
| | $\tilde{w}_0 = 0.25$ | $\mathbf{1.82} \pm 0.05$ | $\mathbf{2.88} \pm 0.20$ | $2.16 \pm 0.07$ | $\mathbf{7.55} \pm 0.07$ | $19.61 \pm 0.09$ |
| | $\tilde{w}_0 = 0.50$ | $0.90 \pm 0.06$ | $2.31 \pm 0.33$ | $0.74 \pm 0.04$ | $4.34 \pm 0.08$ | $19.66 \pm 0.09$ |
| EISEN | NONE | $4.82 \pm 0.25$ | $14.03 \pm 0.62$ | $3.29 \pm 0.20$ | $9.36 \pm 0.14$ | $30.70 \pm 0.07$ |
| | LPROS | $5.87 \pm 0.32$ | $15.15 \pm 0.78$ | $4.11 \pm 0.30$ | $9.32 \pm 0.10$ | $29.69 \pm 0.14$ |
| | HROS-PD | $4.58 \pm 0.25$ | $13.81 \pm 0.62$ | $3.09 \pm 0.27$ | $8.43 \pm 0.15$ | $30.19 \pm 0.07$ |
| | $\tilde{w}_0 = 0.25$ | $\mathbf{7.16} \pm 0.16$ | $11.27 \pm 0.37$ | $\mathbf{6.69} \pm 0.12$ | $\mathbf{12.64} \pm 0.10$ | $30.51 \pm 0.11$ |
| | $\tilde{w}_0 = 0.50$ | $5.92 \pm 0.21$ | $12.79 \pm 0.54$ | $4.55 \pm 0.15$ | $10.91 \pm 0.17$ | $30.73 \pm 0.07$ |
| EXPR | NONE | $5.84 \pm 0.26$ | $13.49 \pm 0.19$ | $4.26 \pm 0.24$ | $10.11 \pm 0.12$ | $30.13 \pm 0.23$ |
| | LPROS | $7.16 \pm 0.26$ | $14.84 \pm 0.73$ | $5.40 \pm 0.22$ | $9.90 \pm 0.17$ | $28.45 \pm 0.13$ |
| | HROS-PD | $6.31 \pm 0.40$ | $14.17 \pm 0.44$ | $4.71 \pm 0.44$ | $10.18 \pm 0.15$ | $30.03 \pm 0.22$ |
| | $\tilde{w}_0 = 0.25$ | $\mathbf{8.50} \pm 0.30$ | $11.92 \pm 0.35$ | $\mathbf{8.10} \pm 0.42$ | $\mathbf{12.50} \pm 0.12$ | $30.30 \pm 0.09$ |
| | $\tilde{w}_0 = 0.50$ | $6.95 \pm 0.33$ | $12.86 \pm 0.46$ | $5.59 \pm 0.34$ | $11.47 \pm 0.07$ | $30.40 \pm 0.09$ |
| GASCH-1 | NONE | $4.98 \pm 0.52$ | $12.56 \pm 1.57$ | $3.64 \pm 0.37$ | $8.27 \pm 0.16$ | $28.44 \pm 0.24$ |
| | LPROS | $5.64 \pm 0.24$ | $13.10 \pm 0.83$ | $4.16 \pm 0.16$ | $8.15 \pm 0.27$ | $26.71 \pm 0.16$ |
| | HROS-PD | $5.63 \pm 0.16$ | $13.36 \pm 0.48$ | $4.13 \pm 0.10$ | $8.27 \pm 0.17$ | $28.28 \pm 0.13$ |
| | $\tilde{w}_0 = 0.25$ | $\mathbf{7.42} \pm 0.36$ | $11.08 \pm 0.50$ | $\mathbf{6.94} \pm 0.36$ | $\mathbf{11.64} \pm 0.20$ | $28.57 \pm 0.22$ |
| | $\tilde{w}_0 = 0.50$ | $6.03 \pm 0.36$ | $12.19 \pm 1.02$ | $4.77 \pm 0.29$ | $10.19 \pm 0.21$ | $28.58 \pm 0.22$ |
| GASCH-2 | NONE | $2.61 \pm 0.12$ | $8.72 \pm 0.61$ | $1.80 \pm 0.08$ | $5.47 \pm 0.13$ | $25.83 \pm 0.06$ |
| | LPROS | $2.86 \pm 0.10$ | $9.92 \pm 0.50$ | $1.98 \pm 0.09$ | $5.49 \pm 0.15$ | $24.97 \pm 0.10$ |
| | HROS-PD | $2.58 \pm 0.10$ | $8.55 \pm 0.22$ | $1.79 \pm 0.06$ | $5.38 \pm 0.13$ | $25.54 \pm 0.14$ |
| | $\tilde{w}_0 = 0.25$ | $\mathbf{4.92} \pm 0.16$ | $9.19 \pm 0.33$ | $\mathbf{4.62} \pm 0.09$ | $\mathbf{10.24} \pm 0.08$ | $25.51 \pm 0.06$ |
| | $\tilde{w}_0 = 0.50$ | $3.47 \pm 0.12$ | $9.21 \pm 0.59$ | $2.63 \pm 0.11$ | $7.66 \pm 0.20$ | $25.76 \pm 0.06$ |
| SEQ | NONE | $4.14 \pm 0.19$ | $11.85 \pm 0.69$ | $2.91 \pm 0.15$ | $9.95 \pm 0.14$ | $29.24 \pm 0.15$ |
| | LPROS | $5.38 \pm 0.16$ | $\mathbf{14.31} \pm 0.44$ | $3.84 \pm 0.12$ | $10.44 \pm 0.16$ | $28.56 \pm 0.27$ |
| | HROS-PD | $4.78 \pm 0.20$ | $13.29 \pm 0.36$ | $3.43 \pm 0.18$ | $10.29 \pm 0.11$ | $29.40 \pm 0.26$ |
| | $\tilde{w}_0 = 0.25$ | $\mathbf{6.60} \pm 0.12$ | $10.57 \pm 0.61$ | $\mathbf{6.29} \pm 0.20$ | $\mathbf{12.13} \pm 0.15$ | $28.91 \pm 0.20$ |
| | $\tilde{w}_0 = 0.50$ | $5.29 \pm 0.11$ | $11.83 \pm 0.63$ | $4.18 \pm 0.15$ | $11.12 \pm 0.20$ | $29.17 \pm 0.16$ |
| SPO | NONE | $0.61 \pm 0.04$ | $3.03 \pm 0.17$ | $0.43 \pm 0.02$ | $3.39 \pm 0.11$ | $21.54 \pm 0.07$ |
| | LPROS | $0.80 \pm 0.04$ | $3.20 \pm 0.18$ | $0.54 \pm 0.03$ | $3.53 \pm 0.08$ | $21.17 \pm 0.12$ |
| | HROS-PD | $0.60 \pm 0.04$ | $3.09 \pm 0.19$ | $0.42 \pm 0.02$ | $3.37 \pm 0.10$ | $21.38 \pm 0.06$ |
| | $\tilde{w}_0 = 0.25$ | $\mathbf{1.94} \pm 0.04$ | $3.21 \pm 0.05$ | $\mathbf{2.17} \pm 0.02$ | $\mathbf{7.82} \pm 0.05$ | $21.21 \pm 0.09$ |
| | $\tilde{w}_0 = 0.50$ | $1.11 \pm 0.04$ | $2.98 \pm 0.11$ | $0.87 \pm 0.03$ | $4.84 \pm 0.07$ | $21.42 \pm 0.08$ |

In Appendix D, attempting to preserve precision, we implement a scheduler that dynamically transforms the weighted objective to unweighted over the course of training. Additionally, we explore the possibility of mixing the weighted and unweighted objectives through linear interpolation. We find that these approaches were mostly unsuccessful in retaining strong $F_1$ while preserving precision. However, in Appendix E, we combine LPROS with weighted loss, and find significant improvements in preserving precision and thus inducing further boosts to the $F_1$ score on most gene product datasets.

### 5.3 Focal Weighting on Gene Product Data

We begin by observing the model performance dynamics as we increase the number of models in our ensemble. For these experiments, as a baseline, we use the imbalance weighted loss, with $\tilde{w}_0 = 0.25$ and all applications of focal weighting are used in tandem with this imbalance weighting. Additionally, predictions for ensembles are made by thresholding the mean of the outputs.

For $F_1$, GMU and bBMA appear to perform well at augmenting recall (although all uncertainty candidate measures appear to outperform the baseline). Consequently, the higher recall and a relatively unchanged precision boost $F_1$, as can be seen in Figure 3a. As one might expect, we observe that the focal loss candidates perform better as the number of heads increases, suggesting that the ensemble becomes a more reliable representation of the model distribution as the size grows. This observation is in contrast to the baseline without focal weighting, which remains relatively independent of ensemble size.

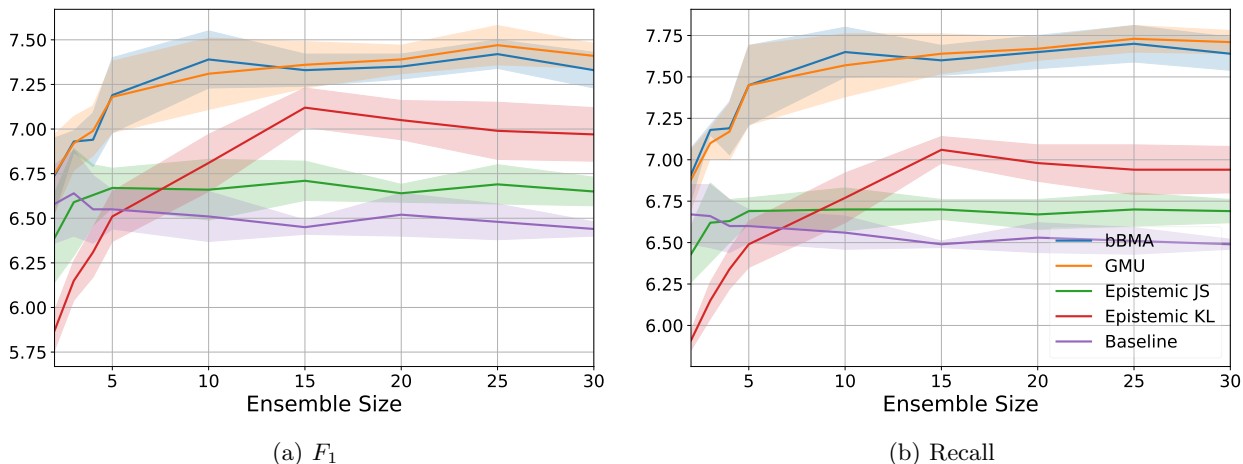

(a) $F_1$                                   (b) Recall

Figure 3: **Focal Weighting with Increasing Ensemble Size on Cellcycle (FUN).** For both displayed measures, the bold central line indicates the mean of the uncertainty term used for focal weighting, while the blurred outline indicates a range of one $\sigma$.

It is nonetheless surprising that epistemic uncertainty underperformed. This may be because, with sufficient training, aleatoric uncertainty typically dominates total uncertainty (Wimmer et al., 2023), leaving the smaller epistemic component with limited impact. Improved model calibration (Wimmer et al., 2023; Pavlovic, 2025) could also yield more robust uncertainty estimates. Alternatively and in contrast to our initial hypothesis, models may benefit from attending to both epistemic and aleatoric uncertainty, especially if neural networks learn weightings more reflective of memorization than generalization.

Hereafter, we implement a static ensemble size of ten to evaluate different kinds of uncertainties for the full catalogue of gene product datasets. We confirm that the same trends we observe in Figure 3 hold true for these other gene product datasets in both FUN and GO. Our results for $F_1$ on FUN are presented in Table 2. A full set of metrics is presented in Appendix B.

Notably, predictions show improvements over even the strongest performing approaches in Table 1. The degree to which improvements result from ensembling is represented via the baseline, which uses only imbalance weighting. There is strong evidence that both bBMA and GMU are practical candidates for focal weighting in improving $F_1$ performance. A deeper look at how the minimum uncertainty gate and focal $k$ impact performance is included in Appendix H.

In addition to gene-product data, and to provide a stronger claim for the generalizability of our proposed approach, additional experiments on "Enron" a text-based dataset, and "Diatoms" an image-based dataset for unicellular algae, are presented in Appendix G. We find that in both, the same trends we observe in gene-product datasets are present. Lastly, compute resource comparisons for the ensemble and a Monte Carlo dropout implementation of our focal weighting are available in Appendix F.

Table 2: **Focal Weighting $F_1$ Score Comparisons (FUN)**. Candidates for uncertainty in focal weighting are compared on the FUN datasets: only imbalance weighting (Baseline), bBMA, GMU, and Epistemic Uncertainty evaluated using JS and KL. Means significantly ($> 2\sigma$) greater than baseline are bolded.

| Method | CELLCYCLE | DERISI | EISEN | EXPR | GASCH-1 | GASCH-2 | SEQ | SPO |
|---|---|---|---|---|---|---|---|---|
| BASELINE | 6.51 ± 0.14 | 2.27 ± 0.04 | 9.31 ± 0.11 | 10.27 ± 0.24 | 9.80 ± 0.06 | 6.47 ± 0.19 | 8.84 ± 0.07 | 2.48 ± 0.03 |
| BBMA | **7.39 ±** 0.16 | **2.54 ±** 0.03 | **9.79 ±** 0.23 | 10.53 ± 0.35 | 10.24 ± 0.30 | **7.23 ±** 0.08 | 8.67 ± 0.12 | **2.91 ±** 0.06 |
| GMU | **7.31 ±** 0.20 | **2.54 ±** 0.03 | **9.78 ±** 0.22 | 10.53 ± 0.30 | 10.35 ± 0.35 | **7.20 ±** 0.07 | 8.65 ± 0.13 | **3.03 ±** 0.02 |
| EPISTEMIC JS | 6.66 ± 0.17 | 2.29 ± 0.03 | 9.52 ± 0.13 | 10.34 ± 0.24 | 9.81 ± 0.06 | 6.52 ± 0.18 | 8.75 ± 0.20 | 2.50 ± 0.03 |
| EPISTEMIC KL | 6.81 ± 0.16 | 2.28 ± 0.02 | 9.39 ± 0.12 | 10.58 ± 0.28 | 9.82 ± 0.07 | 6.35 ± 0.11 | 8.62 ± 0.24 | 2.48 ± 0.04 |

## 5.4 Echinoderm Vision Modelling

For vision models, we experiment with a representative convolutional neural network — ResNet-50 (He et al., 2016). We suspect that CNNs, specialized for visual and spatial data, are less affected by imbalance, at least for simpler tasks, where a task-appropriate encoder is available, or there is abundant data capturing a phenomenon available for training. In other words, settings in which the classification task is easier for the model. Since the approach we introduce is intended for domains where rare node detection is a severe challenge, it follows that the advantages provided would diminish were rare nodes not a serious problem.

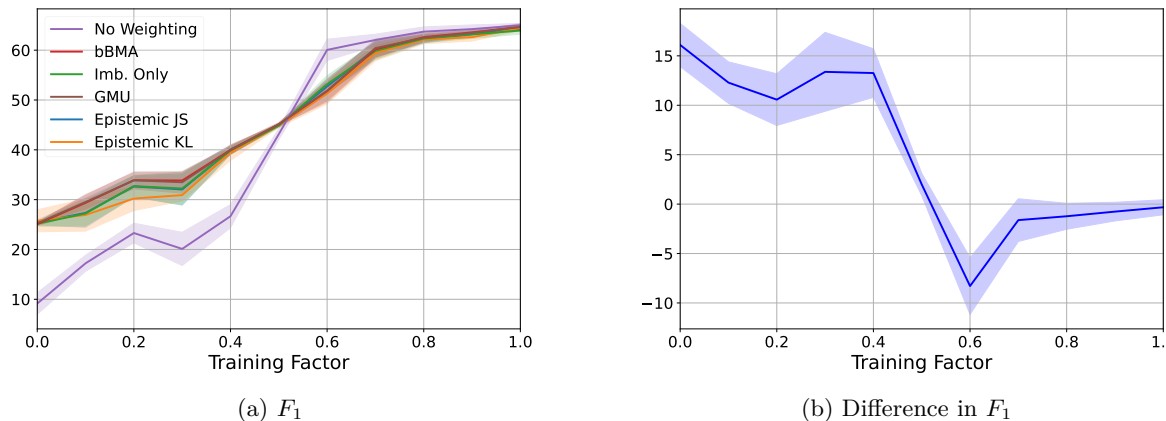

(a) $F_1$             (b) Difference in $F_1$

Figure 4: **Evaluation of $F_1$ over Training Completion on Echinoderms.** The training factor acts as a parameter shifting from 0.0 (randomly initialized encoder) to 1.0 (fully trained ImageNet encoder). In Figure 4a, the weighting methods and an unweighted baseline are compared to each other. In Figure 4b, the difference between GMU and the unweighted baseline is shown, illustrating the dynamical advantage of the weighting method as the encoder becomes better trained.

Figure 4a shows how $F_1$ for echinoderm classification changes as we linearly interpolate between untrained and pre-trained encoders using a "training factor." As training factor increases and noise diminishes, so too do the benefits of imbalance and focal weighting. As with gene product datasets, bBMA and GMU focal-based methods combined with imbalance weighting yield the largest gains when the encoder is noisy. In Appendix I, we provide a deeper look at the individual node performance for the dataset at the extremes with an untrained

and fully-trained encoder. We find that even in the case where the encoder is fully trained, the effects of our method are still present in the rarer nodes. However, the improvements are less dramatic, and these positive effects are approximately cancelled out by small degradations elsewhere in the hierarchy when averaged.

Another way to challenge the model is by limiting the available data for training. We discuss this restricted data experiment in Appendix J. To summarize this experiment, training data is restricted to a range of 1% to 30% of the total available. We find that the relative $F_1$ performances show significant advantages early on with fewer training examples, and it diminishes to around zero when more data becomes available — in support of our hypothesis.

## 6 Conclusion

Our goal was to enable consistent detection of rare nodes deeper in a hierarchy, which are often ignored during model inference due to naturally arising sources of imbalance in datasets — a persistent problem in HML modelling. We attempted to achieve this objective through the implementation of node-wise weighting. We observe significant benefits to the recall of rare nodes in gene product datasets, with increases of up to five times that of existing methods, with models demonstrating the capability of predicting hierarchical nodes which were previously ignored. We also find that focal weighting with a sufficient ensemble size tends to improve recall, especially when focal loss incorporates the bBMA or the GMU as the uncertainty term. In general, the benefits of this approach appear most pronounced for challenging HML tasks.

## 7 Acknowledgements

We thank the Digital Research Alliance of Canada (DRAC) for access to computational resources that supported this research as part of the Resources for Research Groups (RRG) program. Additionally, this research was supported with funding provided by the Ocean Frontier Institute (OFI) as part of the Benthic Ecosystem Mapping and Engagement (BEcoME) project, through an award from the Canada First Research Excellence Fund.

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

## A    Hyperparameters

### A.1    Gene Product Modelling

Our hyperparameters for all methods applied to the gene product datasets are identical to those retrieved from Giunchiglia & Lukasiewicz (2020), to ensure fair comparisons between the approaches. For all our experiments on gene product datasets, we use a batch size of four and set the number of layers equal to three. In other words, we have one input layer mapping from input size to hidden size, one hidden layer mapping from hidden size to the same hidden size, and finally one output layer mapping from hidden size to output size. Additionally, the layers use a dropout of 0.7, and ReLU (Agarap, 2018) as a non-linear activation.

For focal weighting experiments, we use a $U_0 = 0.25$ (matching our applied imbalance minimum gate $\tilde{w}_0$), and a set $k = 1$ — an empirically derived value in preliminary testing.

Table 3: **Hyperparameters for experiments ran on FUN and GO datasets.** Input, Output, and Hidden refer to dimensional size.

| Dataset | Input | Output (FUN) | Output (GO) | Hidden (FUN) | Hidden (GO) | LR (FUN) | LR (GO) | Epochs (FUN/GO) |
|---|---|---|---|---|---|---|---|---|
| Cellcycle | 77 | 499 | 4122 | 500 | 1000 | 1e−4 | 1e−4 | 106 / 62 |
| Derisi | 63 | 499 | 4116 | 500 | 500 | 1e−4 | 1e−4 | 67 / 91 |
| Eisen | 79 | 461 | 3570 | 500 | 500 | 1e−4 | 1e−4 | 110 / 123 |
| Expr | 561 | 499 | 4128 | 1250 | 4000 | 1e−4 | 1e−4 | 20 / 70 |
| Gasch1 | 173 | 499 | 4122 | 1000 | 500 | 1e−4 | 1e−4 | 42 / 122 |
| Gasch2 | 52 | 499 | 4128 | 500 | 500 | 1e−4 | 1e−4 | 123 / 177 |
| Seq | 529 | 499 | 4130 | 2000 | 9000 | 1e−4 | 1e−4 | 13 / 45 |
| Spo | 86 | 499 | 4116 | 250 | 500 | 1e−4 | 1e−4 | 115 / 103 |

### A.2    Echinoderm Modelling

The most drastic difference for our models on echinoderm data is the use of a frozen shared vision encoder. The employed batch size was 128. Given the common encoder, we opt for a multi-layer perceptron classifier component of the same design, consisting of three layers for our ensembles. Dimensional mapping occurs from the embedding dimension of the vision architectures used to the hidden dimension (equal to the embedding dimension), and then to a dimensional size of 13, reflecting the number of nodes in the echinoderm hierarchy. Additionally, we use a learning rate of $1 \times 10^{-3}$ and a weight decay of $1 \times 10^{-4}$, training for 20 epochs.

For our training factor analysis, we apply a $\lambda$ and $1 - \lambda$ mixing between the weights of two encoders: a randomly initialized one and an ImageNet pre-trained one. As we gradually increase the training factor, we run trials using each of the compared methods in accordance with the described settings.

## B    Focal Weighted FUN Results

In Table 4, we present the full results for our experiments with focal weighting on the FUN datasets. We see that while gains in recall and $F_1$ are almost always present for bBMA and "GMU" based uncertainty terms, there are no statistically significant trends for any of the other measures. Notably, we see that even in cases where $F_1$ was not significant, as in "Expr" and "Seq" datasets, the means were still greater. Based on these observations, we believe that including a focal term is a strategy that generates benefits but produces little to no observable downside when appropriate parameters are selected.

## C    Experiments on GO Hierarchy

In Table 5 and Table 6, we present our results for the GO datasets. The overall trend in performance matches those presented for the FUN hierarchy. However, we note the significant drop in HROS-PD performance. This decrease may be expected as the method was not designed for DAG hierarchies (Pereira et al., 2021).

Table 4: **Focal Weighting Comparisons (FUN)**. Different uncertainties for implementing focal weighting are compared for the FUN datasets: only imbalance weighting (Baseline), bBMA, GMU, and Epistemic Uncertainty evaluated using JS and KL. Means which are significantly ($> 2\sigma$) greater than the baseline are bolded.

| Dataset (FUN) | Method | $F_1$ | Precision | Recall | Bin. AP | AP |
|---|---|---|---|---|---|---|
| CELLCYCLE | Baseline | $6.51 \pm 0.14$ | $9.83 \pm 0.31$ | $6.56 \pm 0.10$ | $10.53 \pm 0.11$ | $25.32 \pm 0.03$ |
| | bBMA | $\mathbf{7.39} \pm 0.16$ | $9.47 \pm 0.30$ | $\mathbf{7.65} \pm 0.15$ | $10.46 \pm 0.07$ | $24.79 \pm 0.06$ |
| | GMU | $\mathbf{7.31} \pm 0.20$ | $9.36 \pm 0.36$ | $\mathbf{7.57} \pm 0.19$ | $10.44 \pm 0.10$ | $24.81 \pm 0.07$ |
| | Epistemic JS | $6.66 \pm 0.17$ | $9.83 \pm 0.28$ | $6.70 \pm 0.13$ | $10.58 \pm 0.11$ | $\mathbf{25.43} \pm 0.04$ |
| | Epistemic KL | $6.81 \pm 0.16$ | $10.35 \pm 0.38$ | $6.77 \pm 0.15$ | $10.62 \pm 0.05$ | $\mathbf{25.60} \pm 0.02$ |
| DERISI | Baseline | $2.27 \pm 0.04$ | $3.35 \pm 0.28$ | $2.61 \pm 0.02$ | $7.66 \pm 0.04$ | $19.19 \pm 0.05$ |
| | bBMA | $\mathbf{2.54} \pm 0.03$ | $\mathbf{3.53} \pm 0.04$ | $\mathbf{2.87} \pm 0.01$ | $7.70 \pm 0.03$ | $19.19 \pm 0.05$ |
| | GMU | $\mathbf{2.54} \pm 0.03$ | $\mathbf{3.52} \pm 0.07$ | $\mathbf{2.88} \pm 0.02$ | $7.70 \pm 0.03$ | $19.20 \pm 0.05$ |
| | Epistemic JS | $2.29 \pm 0.03$ | $3.51 \pm 0.21$ | $2.63 \pm 0.02$ | $7.68 \pm 0.04$ | $19.20 \pm 0.05$ |
| | Epistemic KL | $2.28 \pm 0.02$ | $3.52 \pm 0.18$ | $2.61 \pm 0.02$ | $7.70 \pm 0.03$ | $19.19 \pm 0.07$ |
| EISEN | Baseline | $9.31 \pm 0.11$ | $12.87 \pm 0.26$ | $8.85 \pm 0.10$ | $12.57 \pm 0.10$ | $30.22 \pm 0.07$ |
| | bBMA | $\mathbf{9.79} \pm 0.23$ | $12.42 \pm 0.36$ | $\mathbf{9.62} \pm 0.18$ | $12.11 \pm 0.13$ | $29.60 \pm 0.07$ |
| | GMU | $\mathbf{9.78} \pm 0.22$ | $12.50 \pm 0.33$ | $\mathbf{9.59} \pm 0.17$ | $12.16 \pm 0.13$ | $29.65 \pm 0.07$ |
| | Epistemic JS | $9.52 \pm 0.13$ | $13.32 \pm 0.30$ | $8.98 \pm 0.10$ | $12.59 \pm 0.08$ | $30.34 \pm 0.07$ |
| | Epistemic KL | $9.39 \pm 0.12$ | $13.18 \pm 0.34$ | $8.89 \pm 0.10$ | $12.58 \pm 0.06$ | $\mathbf{30.54} \pm 0.06$ |
| EXPR | Baseline | $10.27 \pm 0.24$ | $11.89 \pm 0.17$ | $10.68 \pm 0.30$ | $12.21 \pm 0.09$ | $29.68 \pm 0.07$ |
| | bBMA | $10.53 \pm 0.35$ | $11.36 \pm 0.21$ | $11.53 \pm 0.47$ | $11.91 \pm 0.13$ | $29.02 \pm 0.11$ |
| | GMU | $10.53 \pm 0.30$ | $11.31 \pm 0.23$ | $\mathbf{11.54} \pm 0.39$ | $11.89 \pm 0.08$ | $29.03 \pm 0.11$ |
| | Epistemic JS | $10.34 \pm 0.24$ | $11.82 \pm 0.23$ | $10.79 \pm 0.29$ | $12.25 \pm 0.13$ | $29.76 \pm 0.08$ |
| | Epistemic KL | $10.58 \pm 0.28$ | $11.96 \pm 0.28$ | $11.09 \pm 0.33$ | $12.26 \pm 0.14$ | $29.89 \pm 0.12$ |
| GASCH-1 | Baseline | $9.80 \pm 0.06$ | $12.73 \pm 0.22$ | $9.67 \pm 0.08$ | $11.41 \pm 0.05$ | $27.94 \pm 0.11$ |
| | bBMA | $10.24 \pm 0.30$ | $12.08 \pm 0.51$ | $\mathbf{10.71} \pm 0.18$ | $10.92 \pm 0.10$ | $27.11 \pm 0.12$ |
| | GMU | $10.35 \pm 0.35$ | $12.23 \pm 0.54$ | $\mathbf{10.79} \pm 0.26$ | $10.94 \pm 0.11$ | $27.12 \pm 0.12$ |
| | Epistemic JS | $9.81 \pm 0.06$ | $12.58 \pm 0.21$ | $9.72 \pm 0.04$ | $11.35 \pm 0.03$ | $27.96 \pm 0.11$ |
| | Epistemic KL | $9.82 \pm 0.07$ | $12.60 \pm 0.09$ | $9.70 \pm 0.04$ | $11.34 \pm 0.02$ | $28.02 \pm 0.10$ |
| GASCH-2 | Baseline | $6.47 \pm 0.19$ | $9.70 \pm 0.46$ | $6.42 \pm 0.16$ | $10.56 \pm 0.06$ | $25.54 \pm 0.02$ |
| | bBMA | $\mathbf{7.23} \pm 0.08$ | $9.60 \pm 0.27$ | $\mathbf{7.43} \pm 0.11$ | $10.46 \pm 0.05$ | $25.30 \pm 0.04$ |
| | GMU | $\mathbf{7.20} \pm 0.07$ | $9.59 \pm 0.17$ | $\mathbf{7.37} \pm 0.13$ | $10.45 \pm 0.06$ | $25.32 \pm 0.03$ |
| | Epistemic JS | $6.52 \pm 0.18$ | $9.78 \pm 0.39$ | $6.46 \pm 0.16$ | $10.60 \pm 0.04$ | $\mathbf{25.62} \pm 0.02$ |
| | Epistemic KL | $6.35 \pm 0.11$ | $9.71 \pm 0.45$ | $6.32 \pm 0.09$ | $10.61 \pm 0.09$ | $\mathbf{25.76} \pm 0.02$ |
| SEQ | Baseline | $8.84 \pm 0.07$ | $13.11 \pm 0.28$ | $8.31 \pm 0.07$ | $12.41 \pm 0.11$ | $29.46 \pm 0.10$ |
| | bBMA | $8.67 \pm 0.12$ | $11.70 \pm 0.21$ | $8.56 \pm 0.14$ | $12.02 \pm 0.09$ | $29.23 \pm 0.08$ |
| | GMU | $8.65 \pm 0.13$ | $11.61 \pm 0.36$ | $8.54 \pm 0.12$ | $12.03 \pm 0.13$ | $29.25 \pm 0.10$ |
| | Epistemic JS | $8.75 \pm 0.20$ | $12.69 \pm 0.41$ | $8.32 \pm 0.17$ | $12.39 \pm 0.10$ | $29.59 \pm 0.09$ |
| | Epistemic KL | $8.62 \pm 0.24$ | $12.22 \pm 0.49$ | $8.34 \pm 0.22$ | $12.33 \pm 0.12$ | $\mathbf{29.78} \pm 0.11$ |
| SPO | Baseline | $2.48 \pm 0.03$ | $3.87 \pm 0.31$ | $2.79 \pm 0.02$ | $8.21 \pm 0.06$ | $21.18 \pm 0.02$ |
| | bBMA | $\mathbf{2.91} \pm 0.06$ | $3.78 \pm 0.20$ | $\mathbf{3.20} \pm 0.06$ | $8.19 \pm 0.04$ | $21.16 \pm 0.02$ |
| | GMU | $\mathbf{3.03} \pm 0.02$ | $\mathbf{4.08} \pm 0.06$ | $\mathbf{3.27} \pm 0.03$ | $8.22 \pm 0.03$ | $21.19 \pm 0.02$ |
| | Epistemic JS | $2.50 \pm 0.03$ | $3.86 \pm 0.19$ | $2.80 \pm 0.03$ | $8.21 \pm 0.06$ | $21.24 \pm 0.03$ |
| | Epistemic KL | $2.48 \pm 0.04$ | $3.51 \pm 0.20$ | $2.77 \pm 0.04$ | $8.20 \pm 0.06$ | $\mathbf{21.35} \pm 0.04$ |

## D  Combining Imbalance Weighted Learning Objectives

Given our observations, we are motivated to ask if it is possible to mitigate the decrease in precision by setting $\tilde{w}_0$ to a scheduler. We hypothesized that, as a model learns to recall certain nodes given the weighted learning objective, if we switch to an unweighted learning objective later on in training, once it has already learned to recall certain nodes, precision decline may be mitigated. In testing, we explored both an epoch-based scheduler — where we progressively approach an unweighted learning objective in the later epochs, and a batch-based scheduler — where we approach the unweighted objective every epoch. We find that batch-based

Table 5: **HML Method Comparisons (GO)**. Different strategies in addressing HML imbalance are compared for the FUN datasets: no method used (None), HROS-PD, and node weighting using $\tilde{w}_0$ values of 0.25 and 0.50. Means which are significantly ($> 2\sigma$) greater than all others for a dataset and a measure are bolded.

| Dataset (GO) | Method | $F_1$ | Precision | Recall | Bin. AP | AP |
|---|---|---|---|---|---|---|
| CELLCYCLE | NONE | $1.16 \pm 0.07$ | $3.23 \pm 0.19$ | $0.86 \pm 0.05$ | $16.31 \pm 0.46$ | $41.34 \pm 0.10$ |
| | LPROS | $1.38 \pm 0.05$ | $\mathbf{4.05} \pm 0.13$ | $1.01 \pm 0.04$ | $16.45 \pm 0.44$ | $40.70 \pm 0.14$ |
| | HROS-PD | $0.59 \pm 0.06$ | $2.26 \pm 0.31$ | $0.43 \pm 0.04$ | $11.37 \pm 0.60$ | $40.29 \pm 0.11$ |
| | $\tilde{w}_0 = 0.25$ | $\mathbf{1.94} \pm 0.07$ | $3.22 \pm 0.26$ | $\mathbf{1.86} \pm 0.05$ | $\mathbf{19.84} \pm 0.09$ | $40.93 \pm 0.08$ |
| | $\tilde{w}_0 = 0.50$ | $1.47 \pm 0.04$ | $3.29 \pm 0.14$ | $1.23 \pm 0.04$ | $19.49 \pm 0.14$ | $41.24 \pm 0.12$ |
| DERISI | NONE | $0.34 \pm 0.01$ | $0.69 \pm 0.12$ | $0.30 \pm 0.01$ | $12.88 \pm 0.36$ | $37.01 \pm 0.06$ |
| | LPROS | $0.33 \pm 0.03$ | $0.63 \pm 0.07$ | $0.28 \pm 0.03$ | $12.77 \pm 0.78$ | $37.02 \pm 0.07$ |
| | HROS-PD | $0.19 \pm 0.02$ | $0.55 \pm 0.05$ | $0.15 \pm 0.02$ | $8.79 \pm 0.39$ | $36.38 \pm 0.04$ |
| | $\tilde{w}_0 = 0.25$ | $\mathbf{0.77} \pm 0.05$ | $0.97 \pm 0.22$ | $\mathbf{0.95} \pm 0.04$ | $\mathbf{17.38} \pm 0.08$ | $36.86 \pm 0.05$ |
| | $\tilde{w}_0 = 0.50$ | $0.53 \pm 0.02$ | $0.78 \pm 0.13$ | $0.57 \pm 0.02$ | $17.22 \pm 0.08$ | $36.95 \pm 0.06$ |
| EISEN | NONE | $1.90 \pm 0.18$ | $4.50 \pm 0.32$ | $1.45 \pm 0.15$ | $19.57 \pm 0.61$ | $45.34 \pm 0.15$ |
| | LPROS | $2.15 \pm 0.10$ | $4.90 \pm 0.34$ | $1.62 \pm 0.07$ | $19.25 \pm 0.10$ | $44.70 \pm 0.15$ |
| | HROS-PD | $1.41 \pm 0.09$ | $4.17 \pm 0.19$ | $1.02 \pm 0.07$ | $16.19 \pm 0.51$ | $44.66 \pm 0.16$ |
| | $\tilde{w}_0 = 0.25$ | $\mathbf{3.17} \pm 0.07$ | $4.78 \pm 0.11$ | $\mathbf{3.04} \pm 0.09$ | $\mathbf{22.47} \pm 0.13$ | $44.75 \pm 0.14$ |
| | $\tilde{w}_0 = 0.50$ | $2.40 \pm 0.22$ | $4.40 \pm 0.36$ | $2.04 \pm 0.18$ | $22.38 \pm 0.16$ | $45.14 \pm 0.14$ |
| EXPR | NONE | $3.80 \pm 0.17$ | $7.41 \pm 0.37$ | $2.98 \pm 0.20$ | $16.93 \pm 0.25$ | $38.13 \pm 0.25$ |
| | LPROS | $3.75 \pm 0.05$ | $7.31 \pm 0.24$ | $2.93 \pm 0.07$ | $16.48 \pm 0.25$ | $37.47 \pm 0.30$ |
| | HROS-PD | $2.35 \pm 0.14$ | $5.94 \pm 0.63$ | $1.69 \pm 0.11$ | $15.15 \pm 0.70$ | $38.89 \pm 0.31$ |
| | $\tilde{w}_0 = 0.25$ | $\mathbf{4.61} \pm 0.10$ | $6.87 \pm 0.15$ | $\mathbf{4.13} \pm 0.13$ | $\mathbf{19.37} \pm 0.28$ | $\mathbf{40.41} \pm 0.43$ |
| | $\tilde{w}_0 = 0.50$ | $4.31 \pm 0.09$ | $7.42 \pm 0.20$ | $3.56 \pm 0.12$ | $18.11 \pm 0.23$ | $39.21 \pm 0.18$ |
| GASCH-1 | NONE | $1.52 \pm 0.03$ | $3.08 \pm 0.20$ | $1.19 \pm 0.02$ | $17.79 \pm 0.25$ | $43.61 \pm 0.12$ |
| | LPROS | $1.97 \pm 0.07$ | $4.01 \pm 0.17$ | $1.56 \pm 0.05$ | $18.04 \pm 0.44$ | $43.27 \pm 0.06$ |
| | HROS-PD | $0.86 \pm 0.06$ | $2.38 \pm 0.22$ | $0.63 \pm 0.05$ | $12.96 \pm 0.51$ | $42.48 \pm 0.19$ |
| | $\tilde{w}_0 = 0.25$ | $\mathbf{2.66} \pm 0.08$ | $3.98 \pm 0.25$ | $\mathbf{2.61} \pm 0.05$ | $\mathbf{21.48} \pm 0.09$ | $43.40 \pm 0.11$ |
| | $\tilde{w}_0 = 0.50$ | $2.06 \pm 0.06$ | $3.53 \pm 0.20$ | $1.80 \pm 0.04$ | $21.21 \pm 0.14$ | $43.57 \pm 0.11$ |
| GASCH-2 | NONE | $1.19 \pm 0.02$ | $3.52 \pm 0.10$ | $0.86 \pm 0.02$ | $16.43 \pm 0.49$ | $41.56 \pm 0.10$ |
| | LPROS | $1.30 \pm 0.04$ | $\mathbf{3.80} \pm 0.03$ | $0.95 \pm 0.04$ | $16.54 \pm 0.48$ | $41.45 \pm 0.12$ |
| | HROS-PD | $0.49 \pm 0.03$ | $1.57 \pm 0.10$ | $0.36 \pm 0.02$ | $11.10 \pm 0.57$ | $40.34 \pm 0.10$ |
| | $\tilde{w}_0 = 0.25$ | $\mathbf{1.84} \pm 0.09$ | $3.21 \pm 0.26$ | $\mathbf{1.84} \pm 0.08$ | $20.03 \pm 0.12$ | $41.13 \pm 0.06$ |
| | $\tilde{w}_0 = 0.50$ | $1.42 \pm 0.06$ | $3.42 \pm 0.30$ | $1.20 \pm 0.05$ | $19.82 \pm 0.10$ | $41.40 \pm 0.07$ |
| SEQ | NONE | $3.94 \pm 0.25$ | $8.96 \pm 0.40$ | $2.91 \pm 0.19$ | $17.77 \pm 0.38$ | $38.60 \pm 0.28$ |
| | LPROS | $4.07 \pm 0.14$ | $9.34 \pm 0.38$ | $3.01 \pm 0.10$ | $16.74 \pm 0.59$ | $37.30 \pm 0.81$ |
| | HROS-PD | $2.46 \pm 0.21$ | $6.84 \pm 0.37$ | $1.72 \pm 0.17$ | $15.17 \pm 0.81$ | $39.63 \pm 0.33$ |
| | $\tilde{w}_0 = 0.25$ | $4.46 \pm 0.33$ | $7.93 \pm 0.66$ | $3.65 \pm 0.26$ | $\mathbf{19.69} \pm 0.27$ | $39.90 \pm 0.32$ |
| | $\tilde{w}_0 = 0.50$ | $4.29 \pm 0.21$ | $8.63 \pm 0.23$ | $3.31 \pm 0.19$ | $18.95 \pm 0.37$ | $39.22 \pm 0.44$ |
| SPO | NONE | $0.68 \pm 0.10$ | $1.85 \pm 0.33$ | $0.54 \pm 0.07$ | $13.65 \pm 0.56$ | $38.19 \pm 0.05$ |
| | LPROS | $0.78 \pm 0.09$ | $2.25 \pm 0.42$ | $0.59 \pm 0.05$ | $13.51 \pm 0.27$ | $37.94 \pm 0.06$ |
| | HROS-PD | $0.31 \pm 0.05$ | $0.94 \pm 0.37$ | $0.25 \pm 0.03$ | $9.15 \pm 0.32$ | $36.98 \pm 0.11$ |
| | $\tilde{w}_0 = 0.25$ | $\mathbf{1.26} \pm 0.02$ | $2.29 \pm 0.24$ | $\mathbf{1.29} \pm 0.01$ | $\mathbf{18.12} \pm 0.04$ | $37.88 \pm 0.07$ |
| | $\tilde{w}_0 = 0.50$ | $1.01 \pm 0.02$ | $2.49 \pm 0.11$ | $0.89 \pm 0.02$ | $17.95 \pm 0.20$ | $38.09 \pm 0.05$ |

schedulers perform significantly stronger. While this result isn't surprising from a life-long learning or multi-task learning perspective (Robins, 1995; Silver et al., 2013), it seems to suggest that the loss landscapes for an HML imbalanced weighted objective and an unweighted objective do not easily transition from one to the other and are likely conflicting.

In total, we explore four approaches in combining our learning objectives: linear, exponential, alternating, and mixed loss. Of the four, alternating and mixed loss explore an alternative hypothesis, that the order in which the weighted and unweighted tasks occur does not matter. Alternating refers to switching between the

Table 6: **Focal Weighting Comparisons (GO)**. Different uncertainties for implementing focal weighting are compared for the FUN datasets: only imbalance weighting (Baseline), bBMA, GMU, and Epistemic Uncertainty evaluated using JS and KL. Means which are significantly ($> 2\sigma$) greater than the baseline are bolded.

| Dataset (GO) | Method | $F_1$ | Precision | Recall | Bin. AP | AP |
|---|---|---|---|---|---|---|
| CELLCYCLE | Baseline | $3.50 \pm 0.04$ | $5.26 \pm 0.20$ | $3.24 \pm 0.03$ | $19.71 \pm 0.06$ | $40.39 \pm 0.04$ |
| | bBMA | $\mathbf{3.77} \pm 0.06$ | $5.03 \pm 0.11$ | $\mathbf{3.67} \pm 0.06$ | $18.92 \pm 0.10$ | $39.59 \pm 0.05$ |
| | GMU | $\mathbf{3.79} \pm 0.05$ | $5.04 \pm 0.07$ | $\mathbf{3.69} \pm 0.04$ | $18.89 \pm 0.10$ | $39.56 \pm 0.04$ |
| | Epistemic JS | $3.51 \pm 0.03$ | $5.25 \pm 0.07$ | $3.26 \pm 0.03$ | $19.67 \pm 0.04$ | $40.34 \pm 0.04$ |
| | Epistemic KL | $3.53 \pm 0.04$ | $5.17 \pm 0.19$ | $3.30 \pm 0.03$ | $19.64 \pm 0.09$ | $40.31 \pm 0.07$ |
| DERISI | Baseline | $1.28 \pm 0.04$ | $1.93 \pm 0.12$ | $1.37 \pm 0.04$ | $17.26 \pm 0.01$ | $36.58 \pm 0.02$ |
| | bBMA | $\mathbf{1.40} \pm 0.02$ | $1.97 \pm 0.10$ | $\mathbf{1.51} \pm 0.03$ | $17.03 \pm 0.04$ | $36.38 \pm 0.01$ |
| | GMU | $\mathbf{1.40} \pm 0.02$ | $1.96 \pm 0.10$ | $\mathbf{1.51} \pm 0.01$ | $17.00 \pm 0.03$ | $36.38 \pm 0.01$ |
| | Epistemic JS | $1.27 \pm 0.02$ | $1.81 \pm 0.08$ | $1.37 \pm 0.03$ | $17.24 \pm 0.03$ | $36.56 \pm 0.02$ |
| | Epistemic KL | $1.28 \pm 0.04$ | $1.90 \pm 0.15$ | $1.37 \pm 0.03$ | $17.25 \pm 0.05$ | $36.54 \pm 0.01$ |
| EISEN | Baseline | $5.16 \pm 0.11$ | $7.20 \pm 0.24$ | $4.93 \pm 0.08$ | $22.37 \pm 0.06$ | $45.05 \pm 0.04$ |
| | bBMA | $\mathbf{5.54} \pm 0.12$ | $6.93 \pm 0.22$ | $\mathbf{5.59} \pm 0.15$ | $21.62 \pm 0.11$ | $44.41 \pm 0.09$ |
| | GMU | $\mathbf{5.56} \pm 0.13$ | $6.98 \pm 0.17$ | $\mathbf{5.60} \pm 0.17$ | $21.62 \pm 0.08$ | $44.41 \pm 0.08$ |
| | Epistemic JS | $5.19 \pm 0.12$ | $7.10 \pm 0.23$ | $5.01 \pm 0.14$ | $22.34 \pm 0.09$ | $45.07 \pm 0.04$ |
| | Epistemic KL | $5.18 \pm 0.11$ | $7.09 \pm 0.17$ | $5.00 \pm 0.14$ | $22.34 \pm 0.06$ | $45.09 \pm 0.07$ |
| EXPR | Baseline | $5.28 \pm 0.12$ | $8.67 \pm 0.25$ | $4.50 \pm 0.09$ | $16.34 \pm 0.09$ | $36.13 \pm 0.06$ |
| | bBMA | $5.20 \pm 0.10$ | $8.29 \pm 0.14$ | $4.54 \pm 0.09$ | $16.42 \pm 0.12$ | $35.78 \pm 0.08$ |
| | GMU | $5.26 \pm 0.07$ | $8.28 \pm 0.11$ | $4.57 \pm 0.09$ | $16.51 \pm 0.12$ | $35.84 \pm 0.07$ |
| | Epistemic JS | $5.24 \pm 0.08$ | $8.37 \pm 0.15$ | $4.47 \pm 0.05$ | $16.51 \pm 0.13$ | $35.95 \pm 0.01$ |
| | Epistemic KL | $5.48 \pm 0.11$ | $8.64 \pm 0.22$ | $\mathbf{4.77} \pm 0.07$ | $\mathbf{16.95} \pm 0.13$ | $36.27 \pm 0.09$ |
| GASCH-1 | Baseline | $4.23 \pm 0.07$ | $5.62 \pm 0.10$ | $4.32 \pm 0.05$ | $21.12 \pm 0.03$ | $43.93 \pm 0.02$ |
| | bBMA | $\mathbf{4.92} \pm 0.12$ | $5.90 \pm 0.20$ | $\mathbf{5.33} \pm 0.09$ | $20.28 \pm 0.10$ | $43.16 \pm 0.06$ |
| | GMU | $\mathbf{4.94} \pm 0.11$ | $5.92 \pm 0.21$ | $\mathbf{5.33} \pm 0.12$ | $20.26 \pm 0.07$ | $43.13 \pm 0.08$ |
| | Epistemic JS | $4.30 \pm 0.06$ | $5.63 \pm 0.09$ | $4.43 \pm 0.07$ | $21.09 \pm 0.05$ | $43.92 \pm 0.03$ |
| | Epistemic KL | $4.29 \pm 0.03$ | $5.75 \pm 0.08$ | $\mathbf{4.39} \pm 0.03$ | $21.17 \pm 0.07$ | $43.98 \pm 0.05$ |
| GASCH-2 | Baseline | $3.40 \pm 0.04$ | $5.24 \pm 0.07$ | $3.37 \pm 0.04$ | $20.23 \pm 0.06$ | $41.73 \pm 0.02$ |
| | bBMA | $\mathbf{3.71} \pm 0.05$ | $5.11 \pm 0.14$ | $\mathbf{3.74} \pm 0.05$ | $19.76 \pm 0.07$ | $41.49 \pm 0.04$ |
| | GMU | $\mathbf{3.75} \pm 0.09$ | $5.23 \pm 0.12$ | $\mathbf{3.77} \pm 0.08$ | $19.81 \pm 0.08$ | $41.47 \pm 0.02$ |
| | Epistemic JS | $3.39 \pm 0.09$ | $5.14 \pm 0.13$ | $3.38 \pm 0.08$ | $20.20 \pm 0.08$ | $41.71 \pm 0.02$ |
| | Epistemic KL | $3.35 \pm 0.06$ | $5.07 \pm 0.14$ | $3.31 \pm 0.06$ | $20.23 \pm 0.04$ | $41.70 \pm 0.03$ |
| SEQ | Baseline | $7.52 \pm 0.08$ | $14.03 \pm 0.25$ | $6.04 \pm 0.06$ | $17.77 \pm 0.08$ | $39.39 \pm 0.04$ |
| | bBMA | $7.31 \pm 0.14$ | $13.55 \pm 0.33$ | $5.88 \pm 0.11$ | $17.30 \pm 0.10$ | $38.95 \pm 0.08$ |
| | GMU | $7.43 \pm 0.10$ | $13.80 \pm 0.33$ | $5.98 \pm 0.07$ | $17.28 \pm 0.13$ | $38.91 \pm 0.08$ |
| | Epistemic JS | $7.50 \pm 0.13$ | $14.08 \pm 0.09$ | $6.01 \pm 0.12$ | $17.83 \pm 0.18$ | $39.19 \pm 0.13$ |
| | Epistemic KL | $7.73 \pm 0.12$ | $14.10 \pm 0.31$ | $6.24 \pm 0.10$ | $\mathbf{18.11} \pm 0.05$ | $39.41 \pm 0.10$ |
| SPO | Baseline | $2.01 \pm 0.06$ | $3.36 \pm 0.06$ | $1.91 \pm 0.05$ | $17.91 \pm 0.03$ | $37.57 \pm 0.03$ |
| | bBMA | $\mathbf{2.28} \pm 0.06$ | $3.47 \pm 0.06$ | $\mathbf{2.21} \pm 0.05$ | $17.58 \pm 0.05$ | $37.49 \pm 0.03$ |
| | GMU | $\mathbf{2.34} \pm 0.05$ | $\mathbf{3.54} \pm 0.07$ | $\mathbf{2.26} \pm 0.04$ | $17.62 \pm 0.07$ | $37.49 \pm 0.02$ |
| | Epistemic JS | $2.03 \pm 0.07$ | $3.32 \pm 0.08$ | $1.93 \pm 0.06$ | $17.89 \pm 0.05$ | $37.57 \pm 0.04$ |
| | Epistemic KL | $2.04 \pm 0.08$ | $3.41 \pm 0.15$ | $1.92 \pm 0.06$ | $17.96 \pm 0.08$ | $\mathbf{37.64} \pm 0.02$ |

weighted and unweighted objectives every batch, and mixed loss imposes a $\lambda$ and $1 - \lambda$ relationship between the two objectives.

For our linear scheduler, we evenly space out $\tilde{w}_i$ update steps to the number of training steps ($N_{steps}$) in each epoch. Our exponential scheduler divides exponential steps as $\Delta \tilde{w}_i = \frac{1 - \tilde{w}_i}{(N_{steps} - 1)^k}$. Exponential updates are then performed as $\tilde{w}_{i,t+1} = \tilde{w}_{i,t} + t^k \Delta \tilde{w}_i$, where we introduce the second subscript $t$ to describe the batch index in an epoch, while $k$ is the exponential term. For our experiments, we set $k = 3$. Note that this $k$ is unique to the scheduler and should not be confused with the $k$ applied in focal weighting.

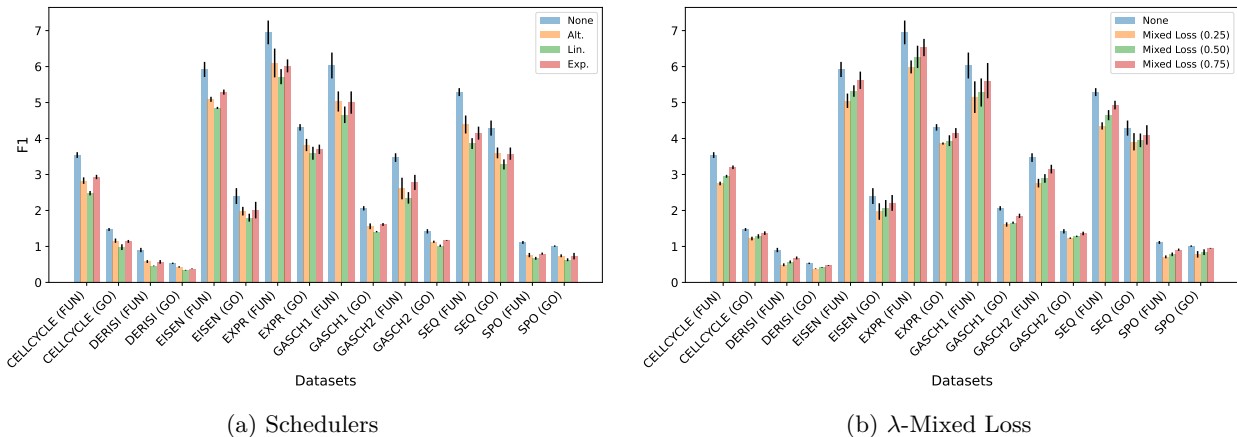

(a) Schedulers                                    (b) $\lambda$-Mixed Loss

Figure 5: **Combining Weighted and Unweighted Objectives.** (a) Scheduler effect on $F_1$ score across all FUN and GO datasets, and (b) mixed loss objective, with varying $\lambda$ effect.

Overall, despite the variety of methods to combine learning objectives, none surpassed results from working solely with the weighted loss. our results are presented in Figure 5.

## E   Combining Imbalance Weighting with LPROS

Table 7: **Combined LPROS and Imbalance Weighting $F_1$ Performance.** Results which exceed 2 $\sigma$ in comparison to the previously best (Prev. Best) performing method for their corresponding dataset are bolded. Previous best values are attained from Table 1 and Table 5.

| Dataset | LPROS + Imb. Weighting | | Prev. Best | |
|---|---|---|---|---|
| | FUN | GO | FUN | GO |
| CELLCYCLE | **5.22** $\pm$ 0.16 | **2.33** $\pm$ 0.18 | 4.85 $\pm$ 0.11 | 1.94 $\pm$ 0.07 |
| DERISI | **2.01** $\pm$ 0.08 | **0.94** $\pm$ 0.05 | 1.82 $\pm$ 0.05 | 0.77 $\pm$ 0.05 |
| EISEN | **8.28** $\pm$ 0.29 | **3.52** $\pm$ 0.12 | 7.16 $\pm$ 0.16 | 3.17 $\pm$ 0.07 |
| EXPR | **9.29** $\pm$ 0.28 | 4.40 $\pm$ 0.11 | 8.50 $\pm$ 0.30 | **4.61** $\pm$ 0.10 |
| GASCH1 | **8.05** $\pm$ 0.16 | **3.09** $\pm$ 0.04 | 7.42 $\pm$ 0.36 | 2.66 $\pm$ 0.08 |
| GASCH2 | **5.39** $\pm$ 0.22 | **2.09** $\pm$ 0.05 | 4.92 $\pm$ 0.16 | 1.84 $\pm$ 0.09 |
| SEQ | **7.57** $\pm$ 0.35 | 4.60 $\pm$ 0.14 | 6.60 $\pm$ 0.12 | 4.46 $\pm$ 0.33 |
| SPO | **2.24** $\pm$ 0.03 | 1.35 $\pm$ 0.08 | 1.94 $\pm$ 0.04 | 1.26 $\pm$ 0.02 |

Rather than attempting to merge losses to enhance precision in imbalance weighted learning as in Appendix D, we could seek to integrate resampling methods instead. In Table 1 and Table 5, we observe that LPROS often attains the highest precision in comparison to other approaches. Furthermore, LPROS may be applied in conjunction with DAG-like hierarchies, as in the case of our GO datasets, in contrast to HROS-PD. We are therefore motivated to ask if this performance could be combined constructively with the effects of imbalance weighting.

Firstly, examining the individual node performances across datasets revealed that LPROS impacts the nodes in a multi-faceted manner. While some nodes traded recall for improved precision, others saw precision gains. Additionally, the LPROS-trained model began predicting previously unpredicted nodes, but also ceased predictions for some previously predicted nodes. These results suggest that LPROS behaves differently from imbalance weighting, and their combination may derive synergistic benefits. For the purposes of this work, and given finite computational budgets, we calculate node weights after resampling. We reason that this ordering is a safer option to provide beneficial results, given the unknown effects of their combination. Our results are presented in Table 7.

As observed, the combined approach exceeds the previous best method (imbalance weighting with $\tilde{w}_0 = 0.25$) across almost all gene product datasets. Although we had expected a jump in $F_1$ score due to increased precision, which was indeed observed, surprisingly, the recall too was boosted as a result of this combination. However, we caution that the combined method did not perform strongly on our echinoderm dataset. It is difficult to ascertain why this unexpected difference in performance occurred, as there may be a myriad of issues pertaining to computer vision, such as hyperparameter specifications for differing architectures, or augmentation-related influences, and so on. Further investigation would be required.

## F  Dropout and Ensemble Computational Performance

To examine the viability of computationally faster alternatives to full ensembles or classifier ensembles, we test a dropout-based approach for our focal weighting scheme, using GMU. We present our findings in Table 8.

Overall, switching from ensemble to dropout results in a reduction in training time of $68.1 \pm 5.2\%$, a reduction in memory usage of $66.4 \pm 13.2\%$, and a reduction in $F_1$ score of $6.06 \pm 5.48\%$. Interestingly, we observe an increase in $F_1$ performance for the SEQ dataset. Generally, however, switching to $F_1$ appears to drop the performance to slightly higher than the results from Table 4 for the imbalance-only weighting.

Dropout was implemented by passing the data through the model a number of times equal to the number of models present in the ensemble during training. Our training hyperparameters are unchanged from Appendix A. No dropout is used during testing and inference, including for the reported results, as dropout was used purely as a means to mimic the hypothetical distribution of plausible models to capture uncertainty during training.

Table 8: **Comparisons of Ensemble and Dropout on FUN Datasets**. Dropout performance is presented in the top row for each dataset, while ensemble performance is presented in the bottom row.

| Dataset (FUN) | Inference (s/epoch) | it/s | Mem. Usage (MB) | $F_1$ |
|---|---|---|---|---|
| CELLCYCLE | $1.99 \pm 0.07$ | $434.0 \pm 14.4$ | 47.2 | 6.65 |
| | $5.55 \pm 0.06$ | $155.0 \pm 1.7$ | 121.4 | 7.46 |
| DERISI | $1.97 \pm 0.08$ | $432.2 \pm 15.9$ | 46.4 | 2.23 |
| | $5.50 \pm 0.07$ | $154.5 \pm 2.0$ | 119.6 | 2.47 |
| EISEN | $1.20 \pm 0.06$ | $471.3 \pm 20.4$ | 38.6 | 9.35 |
| | $3.54 \pm 0.04$ | $159.6 \pm 1.6$ | 110.3 | 9.90 |
| EXPR | $2.07 \pm 0.10$ | $418.8 \pm 19.2$ | 101.9 | 10.03 |
| | $7.91 \pm 0.08$ | $109.3 \pm 1.07$ | 503.7 | 10.58 |
| GASCH-1 | $2.01 \pm 0.09$ | $430.1 \pm 16.7$ | 68.5 | 10.02 |
| | $6.61 \pm 0.06$ | $130.5 \pm 1.2$ | 300.9 | 10.20 |
| GASCH-2 | $1.95 \pm 0.07$ | $442.9 \pm 14.5$ | 46.2 | 6.57 |
| | $5.52 \pm 0.09$ | $156.4 \pm 2.6$ | 118.7 | 7.21 |
| SEQ | $2.37 \pm 0.09$ | $375.3 \pm 13.8$ | 152.2 | 9.19 |
| | $11.13 \pm 0.10$ | $80.0 \pm 0.7$ | 1015.9 | 8.69 |
| SPO | $1.94 \pm 0.07$ | $438.5 \pm 16.1$ | 42.2 | 2.53 |
| | $5.32 \pm 0.05$ | $159.6 \pm 1.6$ | 71.0 | 2.88 |

These tests were conducted on a workstation with an i9-13900KF CPU, Nvidia RTX 4090 GPU, and 32 GB of RAM.

## G  Additional Datasets

We expand upon the data modes explored using FUN and GO gene product data. To this end, we apply our experiments to two additional datasets: "Enron" (Klimt & Yang, 2004) and "Diatom" (Dimitrovski et al., 2011). Enron is a language dataset, consisting of emails, each of input size 1000, with 56 nodes, while Diatom

encompasses unicellular micro-algae processed image data, each of input size 371 with 398 nodes (Wehrmann et al., 2018).

Our results are provided in Table 9. We observe that the benefits of imbalance and focal loss appear to be definitive in Enron, while the benefits appear more muted in Diatom, with imbalance weighting alone performing slightly worse than the no weighting baseline, in terms of mean performance alone.

Table 9: **Focal Weighting Comparisons (Others)**. Different uncertainties for implementing focal weighting are compared for the additional datasets: "No Weighting" (no imbalance or focal weighting applied), "Imb. Only" (imbalance weighting applied only), bBMA, GMU, and Epistemic Uncertainty were evaluated using JS and KL. Means which are significantly ($> 2\sigma$) greater than the "No Weighting" and "Imb. Only" baselines are bolded.

| Dataset (FUN) | Method | $F_1$ | Precision | Recall | Bin. AP | AP |
|---|---|---|---|---|---|---|
| ENRON | NO WEIGHTING | $12.65 \pm 0.11$ | $18.23 \pm 0.94$ | $11.30 \pm 0.09$ | $47.80 \pm 0.17$ | $75.77 \pm 0.05$ |
| | IMB. ONLY | $14.64 \pm 0.09$ | $18.53 \pm 1.24$ | $14.37 \pm 0.04$ | $50.60 \pm 0.13$ | $75.86 \pm 0.04$ |
| | bBMA | $\mathbf{15.39} \pm 0.06$ | $\mathbf{20.07} \pm 0.06$ | $\mathbf{14.87} \pm 0.07$ | $50.69 \pm 0.16$ | $75.90 \pm 0.06$ |
| | GMU | $\mathbf{15.40} \pm 0.08$ | $\mathbf{20.21} \pm 0.21$ | $\mathbf{14.85} \pm 0.05$ | $50.71 \pm 0.23$ | $75.88 \pm 0.06$ |
| | EPISTEMIC JS | $14.68 \pm 0.11$ | $18.78 \pm 1.61$ | $14.41 \pm 0.05$ | $50.61 \pm 0.18$ | $75.88 \pm 0.05$ |
| | EPISTEMIC KL | $14.84 \pm 0.13$ | $18.92 \pm 1.12$ | $\mathbf{14.56} \pm 0.09$ | $50.82 \pm 0.24$ | $75.88 \pm 0.05$ |
| DIATOM | NO WEIGHTING | $57.47 \pm 0.36$ | $64.82 \pm 0.32$ | $55.12 \pm 0.40$ | $57.53 \pm 0.26$ | $79.43 \pm 0.04$ |
| | IMB. ONLY | $57.17 \pm 0.39$ | $63.66 \pm 0.49$ | $55.39 \pm 0.35$ | $56.69 \pm 0.28$ | $79.01 \pm 0.02$ |
| | bBMA | $57.55 \pm 0.26$ | $63.99 \pm 0.34$ | $55.83 \pm 0.23$ | $57.39 \pm 0.31$ | $79.24 \pm 0.02$ |
| | GMU | $57.65 \pm 0.21$ | $64.05 \pm 0.30$ | $\mathbf{55.99} \pm 0.15$ | $57.34 \pm 0.22$ | $79.26 \pm 0.02$ |
| | EPISTEMIC JS | $57.12 \pm 0.16$ | $63.54 \pm 0.20$ | $55.38 \pm 0.20$ | $56.89 \pm 0.20$ | $79.09 \pm 0.02$ |
| | EPISTEMIC KL | $57.12 \pm 0.16$ | $63.35 \pm 0.26$ | $55.57 \pm 0.21$ | $56.59 \pm 0.28$ | $79.05 \pm 0.04$ |

# H Exploring Robustness of $\tilde{w}_0$, $u_0$, and focal $k$

When designing the information gates, $\tilde{w}_0$ and $u_0$ (not to be confused with its tensor form $U_0$), we aimed to introduce parameters which did not require meticulous fine-tuning for each dataset, as such a requirement would diminish its value. For the main experiments, a value of 0.25 was used for both gates. To investigate the effect of these gates at a greater resolution, we conduct an experiment on the Expr (FUN) dataset, testing a variety of values ranging from 0 to 1.5. Additionally, we test the focal exponent $k$ from a range of 0 to 3.5.

As can be seen in Figure 6a, we observe that as $\tilde{w}_0$ increases, the precision increases logarithmically, while recall decays exponentially, both with heteroscedasticity. This behaviour is likely why we are able to significantly increase recall with relatively small costs to precision. We also find that the optimal $F_1$ in this case is in the values lower than $\tilde{w}_0 = 0.2$, which is smaller than our general setting in this work. Nonetheless, this is a parameter that should be chosen for the particular problem of interest. If precision is a factor that cannot be compromised, higher values of around 0.5 to 0.8 appear reasonable, and may still deliver strong recall — as we demonstrate in our earlier results. However, if detecting rare cases is especially important, smaller values that more generally improve $F_1$ may be desirable. In contrast, the $u_0$ and $k$ have a dampened effect. We note that $u_0$ decreases recall for values larger than 0.1 to 0.2, while gradually increasing precision. Additionally, a larger $k$ tends to decrease precision and increase recall. The former likely acts similarly to $\tilde{w}_0$, in that it is a similar minimum gating term, and perhaps because the latter emphasizes extremes, larger values benefit recall.

# I Effects on BenthicNet-E Nodes

In this section, we examine the effects of the combined weighted loss more closely on BenthicNet-E. We present our findings in Figure 7.

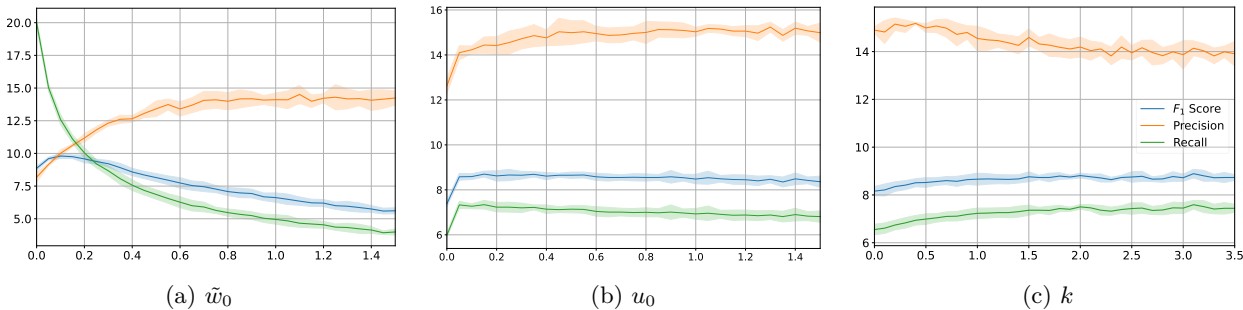

(a) $\tilde{w}_0$        (b) $u_0$        (c) $k$

Figure 6: **Exploring Weighted and Focal Hyperparameters' Effect on Performance.** Plotted is a representative example of the effect of $\tilde{w}_0$, $u_0$, and $k$ on model $F_1$, precision, and recall performance using the Expr (FUN) dataset. The results for $u_0$, and $k$ were attained with $\tilde{w}_0$ set to 0.25.

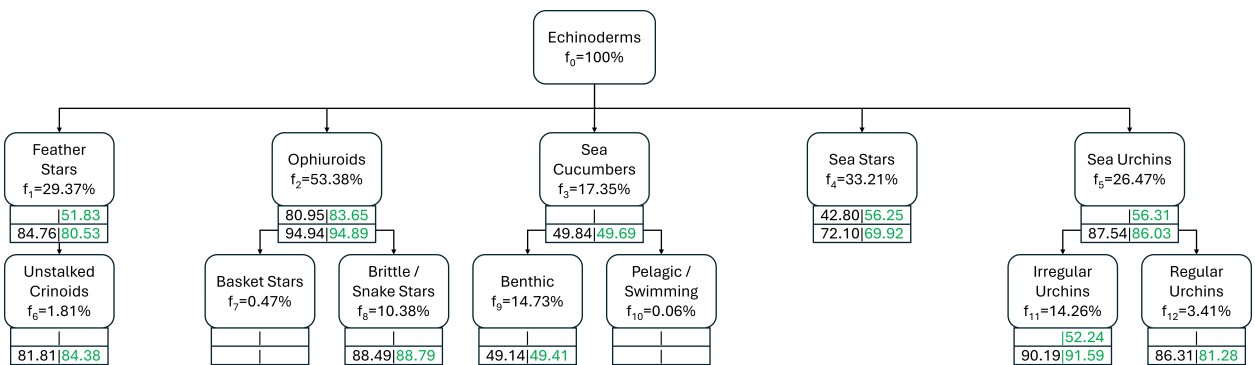

Figure 7: **Node-level Examination for BenthicNet-E Performance.** Displayed is the BenthicNet-E hierarchy. The $F_1$ scores attained are shown below each node. The top row in the score boxes refers to the untrained encoder extreme, while the lower row shows the fully trained case. The left of the score boxes shows the unweighted performance, while the right shows the performance for a combined imbalance and focal weighted objective.

Despite its apparently limited impact in the case of a fully trained encoder, we observe the more subtle effects present. Notably, the weighted approach appears to increase performance for deeper nodes, with a slight cost to nodes higher in the hierarchy — as we might expect. As these deeper nodes tend to have greater weightings, due to imbalance and their difficulty to learn.

In the untrained case, the improvement is drastic, as we observe not only performance gains in the deeper nodes, but also for the higher-level nodes, with multiple previously undetected nodes now becoming detectable.

In general, we posit that the method does not fail in these cases for trained or untrained encoders. Rather, we interpret this observation as evidence that it remains active in its intended purpose of improving detection and performance on rare nodes. The difference between the two cases is that, if the model is learned to a degree that it largely already recognizes the rare nodes present in a dataset, then there is little more to gain on these rare nodes. Therefore, there may exist disruptions on the common nodes as we reshuffle the weightings to prioritize learning the rare nodes.

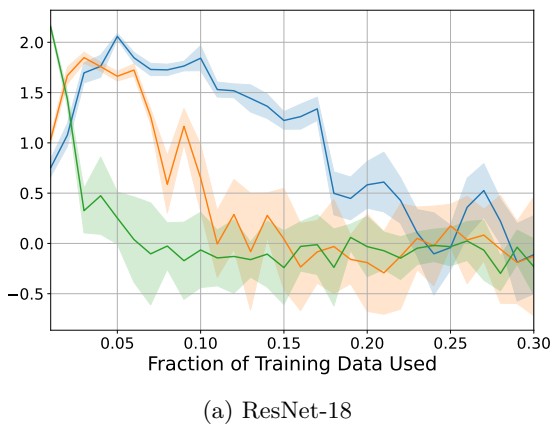

(a) ResNet-18

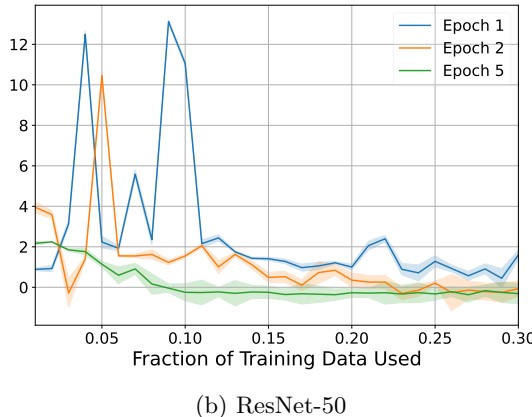

(b) ResNet-50

Figure 8: **Weighted vs. Unweighted $F_1$ Advantage by Training Fraction.** The y-axis advantage factor is proportional to the percent increase in $F_1$ due to weighting: 2.0 means 200% higher, 1.0 is 100% higher, 0.0 is equal. Trendlines are displayed for training on available data for 1, 2, and 5 epochs.

## J   Data Ablation Study for BenthicNet-E

An alternative way of controlling the difficulty of a classification task is by limiting the amount of available data for a model during training. By reducing the training data, we expect our approach in combining imbalance and uncertainty weightings to greatly benefit the model in its downstream task when available data and training are low. As we increase available data, we expect the advantage from our methods to diminish as available data saturates the model to the phenomenon and rare node detection becomes less of a challenge.

In this ablation study, our training parameters are largely the same as our main experiments. However, batch size has been altered to $\min(d, 128)$, where $d$ is the dataset length. Additionally, the encoder and classifiers are prepared from scratch with all weights trainable. We compare the advantage in $F_1$ score between the full set of weightings applied and the unweighted baseline. The uncertainty method applied is GMU. Advantage is calculated as $\text{Adv.} = \frac{F_1^w - F_1^u}{F_1^u}$; where $F_1^w$ is the $F_1$ score with weighting, and $F_1^u$ is the $F_1$ score without weighting. Figure 8 displays our results.

Notably, our results provide additional evidence for the hypothesis regarding where our method provides the greatest advantages. It is also worth noting that the larger ResNet-50 backbone develops greater instability for these low data domains, in comparison to the ResNet-18 architecture. This effect may be the result of overfitting. If that is the case, the emphasis on rare nodes may help combat overfitting, as can be seen by the dramatic improvements in $F_1$ relative to the baseline in these settings.

