|---|---|---|---|---|---|---|
| CELLCYCLE | None | 2.61 ± 0.09 | 9.81 ± 0.70 | 1.74 ± 0.06 | 6.26 ± 0.11 | 25.68 ± 0.13 |
| | LPROS | 2.98 ± 0.18 | 9.70 ± 0.80 | 2.13 ± 0.19 | 6.42 ± 0.10 | 24.81 ± 0.25 |
| | HROS-PD | 2.70 ± 0.25 | 9.57 ± 0.90 | 1.88 ± 0.22 | 6.18 ± 0.12 | 25.57 ± 0.16 |
| | $\tilde{w}_0 = 0.25$ | **4.85** ± 0.11 | 9.13 ± 0.36 | **4.59** ± 0.11 | **10.21** ± 0.08 | 25.45 ± 0.10 |
| | $\tilde{w}_0 = 0.50$ | 3.54 ± 0.08 | 9.34 ± 0.36 | 2.68 ± 0.07 | 8.20 ± 0.08 | 25.69 ± 0.12 |
| DERISI | None | 0.46 ± 0.03 | 1.75 ± 0.21 | 0.34 ± 0.03 | 2.67 ± 0.08 | 19.58 ± 0.10 |
| | LPROS | 0.50 ± 0.02 | 1.96 ± 0.16 | 0.35 ± 0.02 | 2.66 ± 0.08 | 19.18 ± 0.10 |
| | HROS-PD | 0.44 ± 0.02 | 1.82 ± 0.41 | 0.33 ± 0.01 | 2.70 ± 0.09 | 19.39 ± 0.11 |
| | $\tilde{w}_0 = 0.25$ | **1.82** ± 0.05 | **2.88** ± 0.20 | 2.16 ± 0.07 | **7.55** ± 0.07 | 19.61 ± 0.09 |
| | $\tilde{w}_0 = 0.50$ | 0.90 ± 0.06 | 2.31 ± 0.33 | 0.74 ± 0.04 | 4.34 ± 0.08 | 19.66 ± 0.09 |
| EISEN | None | 4.82 ± 0.25 | 14.03 ± 0.62 | 3.29 ± 0.20 | 9.36 ± 0.14 | 30.70 ± 0.07 |
| | LPROS | 5.87 ± 0.32 | 15.15 ± 0.78 | 4.11 ± 0.30 | 9.32 ± 0.10 | 29.69 ± 0.14 |
| | HROS-PD | 4.58 ± 0.25 | 13.81 ± 0.62 | 3.09 ± 0.27 | 8.43 ± 0.15 | 30.19 ± 0.07 |
| | $\tilde{w}_0 = 0.25$ | **7.16** ± 0.16 | 11.27 ± 0.37 | **6.69** ± 0.12 | **12.64** ± 0.10 | 30.51 ± 0.11 |
| | $\tilde{w}_0 = 0.50$ | 5.92 ± 0.21 | 12.79 ± 0.54 | 4.55 ± 0.15 | 10.91 ± 0.17 | 30.73 ± 0.07 |
| EXPR | None | 5.84 ± 0.26 | 13.49 ± 0.19 | 4.26 ± 0.24 | 10.11 ± 0.12 | 30.13 ± 0.23 |
| | LPROS | 7.16 ± 0.26 | 14.84 ± 0.73 | 5.40 ± 0.22 | 9.90 ± 0.17 | 28.45 ± 0.13 |
| | HROS-PD | 6.31 ± 0.40 | 14.17 ± 0.44 | 4.71 ± 0.44 | 10.18 ± 0.15 | 30.03 ± 0.22 |
| | $\tilde{w}_0 = 0.25$ | **8.50** ± 0.30 | 11.92 ± 0.35 | **8.10** ± 0.42 | **12.50** ± 0.12 | 30.30 ± 0.09 |
| | $\tilde{w}_0 = 0.50$ | 6.95 ± 0.33 | 12.86 ± 0.46 | 5.59 ± 0.34 | 11.47 ± 0.07 | 30.40 ± 0.09 |
| GASCH-1 | None | 4.98 ± 0.52 | 12.56 ± 1.57 | 3.64 ± 0.37 | 8.27 ± 0.16 | 28.44 ± 0.24 |
| | LPROS | 5.64 ± 0.24 | 13.10 ± 0.83 | 4.16 ± 0.16 | 8.15 ± 0.27 | 26.71 ± 0.16 |
| | HROS-PD | 5.63 ± 0.16 | 13.36 ± 0.48 | 4.13 ± 0.10 | 8.27 ± 0.17 | 28.28 ± 0.13 |
| | $\tilde{w}_0 = 0.25$ | **7.42** ± 0.36 | 11.08 ± 0.50 | **6.94** ± 0.36 | **11.64** ± 0.20 | 28.57 ± 0.22 |
| | $\tilde{w}_0 = 0.50$ | 6.03 ± 0.36 | 12.19 ± 1.02 | 4.77 ± 0.29 | 10.19 ± 0.21 | 28.58 ± 0.22 |
| GASCH-2 | None | 2.61 ± 0.12 | 8.72 ± 0.61 | 1.80 ± 0.08 | 5.47 ± 0.13 | 25.83 ± 0.06 |
| | LPROS | 2.86 ± 0.10 | 9.92 ± 0.50 | 1.98 ± 0.09 | 5.49 ± 0.15 | 24.97 ± 0.10 |
| | HROS-PD | 2.58 ± 0.10 | 8.55 ± 0.22 | 1.79 ± 0.06 | 5.38 ± 0.13 | 25.54 ± 0.14 |
| | $\tilde{w}_0 = 0.25$ | **4.92** ± 0.16 | 9.19 ± 0.33 | **4.62** ± 0.09 | **10.24** ± 0.08 | 25.51 ± 0.06 |
| | $\tilde{w}_0 = 0.50$ | 3.47 ± 0.12 | 9.21 ± 0.59 | 2.63 ± 0.11 | 7.66 ± 0.20 | 25.76 ± 0.06 |
| SEQ | None | 4.14 ± 0.19 | 11.85 ± 0.69 | 2.91 ± 0.15 | 9.95 ± 0.14 | 29.24 ± 0.15 |
| | LPROS | 5.38 ± 0.16 | **14.31** ± 0.44 | 3.84 ± 0.12 | 10.44 ± 0.16 | 28.56 ± 0.27 |
| | HROS-PD | 4.78 ± 0.20 | 13.29 ± 0.36 | 3.43 ± 0.18 | 10.29 ± 0.11 | 29.40 ± 0.26 |
| | $\tilde{w}_0 = 0.25$ | **6.60** ± 0.12 | 10.57 ± 0.61 | **6.29** ± 0.20 | **12.13** ± 0.15 | 28.91 ± 0.20 |
| | $\tilde{w}_0 = 0.50$ | 5.29 ± 0.11 | 11.83 ± 0.63 | 4.18 ± 0.15 | 11.12 ± 0.20 | 29.17 ± 0.16 |
| SPO | None | 0.61 ± 0.04 | 3.03 ± 0.17 | 0.43 ± 0.02 | 3.39 ± 0.11 | 21.54 ± 0.07 |
| | LPROS | 0.80 ± 0.04 | 3.20 ± 0.18 | 0.54 ± 0.03 | 3.53 ± 0.08 | 21.17 ± 0.12 |
| | HROS-PD | 0.60 ± 0.04 | 3.09 ± 0.19 | 0.42 ± 0.02 | 3.37 ± 0.10 | 21.38 ± 0.06 |
| | $\tilde{w}_0 = 0.25$ | **1.94** ± 0.04 | 3.21 ± 0.05 | **2.17** ± 0.02 | **7.82** ± 0.05 | 21.21 ± 0.09 |
| | $\tilde{w}_0 = 0.50$ | 1.11 ± 0.04 | 2.98 ± 0.11 | 0.87 ± 0.03 | 4.84 ± 0.07 | 21.42 ± 0.08 |

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

 Adv. $= \frac{F_1^{\text{w}} - F_1^{\text{u}}}{F_1^{\text{u}}}$; where $F_1^{\text{w}}$ is the F1 score with weighting, and $F_1^{\text{u}}$ is the F1 score without weighting. Figure 8 displays our results.

Notably, our results provide additional evidence for the hypothesis regarding where our method provides the greatest advantages. It is also worth noting that the larger ResNet-50 backbone develops greater instability for these low data domains, in comparison to the ResNet-18 architecture. This effect may be the result of overfitting. If that is the case, the emphasis on rare nodes may help combat overfitting, as can be seen by the dramatic improvements in F1 relative to the baseline in these settings.