# OpenReview forum: "Improving Detection of Rare Nodes in Hierarchical Multi-Label Learning"
_TMLR — Accepted by TMLR_

### Review · Reviewer_d5mR · 2025-10-10

**Summary Of Contributions:**

The paper addresses the Rare Node Problem in Hierarchical Multi-Label Classification (HMLC), where models fail to achieve satisfactory Recall for deep, low-frequency categories. The core contribution is a novel weighted loss objective, L_focal, which multiplies a Node-Wise Imbalance Weight (W) by an Ensemble-based Focal Weight (U) applied to the baseline LMC loss.

Key Strengths:

Conceptual Synergy: The principle of combining statistical rarity and model confusion (epistemic uncertainty) into a single multiplicative penalty is a novel and powerful mechanism for gradient prioritization.

Exceptional Empirical Gain: The reported up to five-fold increase in Recall and statistically significant F1 score gains on genomic benchmarks provide strong empirical validation that the core thesis is correct.

Key Weaknesses:

Computational Infeasibility: The reliance on an explicit model ensemble for U creates a massive cost and memory barrier.

Generalization Fragility: The method's failure to perform strongly on the vision dataset (BenthicNet-E) suggests a lack of robustness across data modalities.

**Audience:**

Yes

**Audience Explanation:**

The findings are of high interest to the TMLR audience because they solve a major, practical, and long-standing failure mode in HMLC. HMLC is a critical component in domains such as bioinformatics (Gene Ontology classification), large-scale document organization, and taxonomy construction. The ability to achieve reliable deep-level classification directly impacts scientific discovery and knowledge organization. The paper provides a novel principle that can initiate further research into low-cost, uncertainty-aware loss functions.

**Broader Impact Concerns:**

No significant broader impact concerns are present beyond the standard risks associated with machine learning (e.g., bias propagation), which is mitigated by the objective nature of the genomic and taxonomic data used. No additional statement is required.

**Claims And Evidence:**

Yes

**Claims Explanation:**

The claims are supported by convincing empirical evidence derived from rigorous ablation studies on established benchmark datasets (e.g., Gene Ontology). The reported gains in both Recall (up to 5×) and statistically significant F1 score improvements validate that the proposed L_focal objective successfully directs model capacity toward the detection of rare nodes. Furthermore, the systematic testing against baseline methods (LMC and resampling methods like LPROS) clearly establishes the superiority of the combined weighting approach. The LMC component ensures structural coherence (the hierarchy constraint) is maintained, which is a key technical requirement.

**Requested Changes:**

1. Address Computational Infeasibility (Ensemble Cost)

The current reliance on an explicit model ensemble for computing $U$ introduces substantial computational overhead, undermining the method’s practical feasibility. The authors must:

Explore and quantify the performance of at least one low-cost proxy for model uncertainty (e.g., Monte Carlo Dropout, Evidential Deep Learning) to demonstrate a pathway toward a single-model, scalable solution.

Include a transparent cost–benefit analysis (e.g., a table comparing F1 score vs. inference latency / memory overhead) to clearly quantify the computational trade-off introduced by the ensemble.

2. Address Generalization Fragility (Vision Data Failure)

The method’s failure on the BenthicNet-E vision dataset requires a deeper, mechanism-focused explanation rather than surface-level reporting. Specifically:

Discuss the interaction between the highly weighted semantic loss, CNN gradient dynamics, and augmentation strategies.

Identify the failure modes responsible for the degradation in visual domains.

Suggest explicit boundary conditions under which the proposed method is expected to perform reliably (e.g., tasks involving semantic or discrete hierarchies).

3. Address Optimization Stability ($\tilde{w}_0$ Sensitivity)

The hyperparameter $\tilde{w}_0$ currently serves as a manual control over the precision–recall trade-off, which raises concerns about optimization stability. The authors must:

Provide a rigorous discussion of $\tilde{w}_0$ as a sensitivity factor and its impact on convergence.

Explore methods to dynamically learn or adapt $\tilde{w}_0$ during training, or

Present empirical evidence that the objective function remains stable across a broad, practically relevant range of $\tilde{w}_0$ values.

Strengthening Revisions (Recommended)
1. Wider Domain Testing

Evaluate the proposed method on an additional, distinct data modality (e.g., text classification hierarchy) to provide stronger evidence of generalizability beyond the current genomic datasets.

2. Detailed Ablation Analysis

Include a supplementary ablation study focusing on the performance of common (frequent) nodes to confirm that the significant recall gains observed on rare nodes do not cause catastrophic underfitting or reduced predictive quality on other parts of the hierarchy.

3. Reproducibility Documentation

Explicitly state recommended initialization or starting ranges for key hyperparameters ($U_0$, $k$, $\tilde{w}_0$) to lower the barrier for independent reproduction and reliability validation.

Summary Justification

These revisions are critical to ensure the proposed approach is computationally feasible, theoretically robust, and empirically generalizable across domains. The current formulation demonstrates promising results but lacks practical viability and clarity on optimization stability. Addressing the above points would significantly strengthen both the methodological soundness and the real-world impact of this work.

---

> ### Author Response · Authors · 2025-11-25
> **Rebuttal for Reviewer d5mR**
>
> Thank you for the valuable feedback and suggestions. We address each of your points below.
>
> 1. Address Computational Infeasibility (Ensemble Cost). The current reliance on an explicit model ensemble for computing $U$ introduces substantial computational overhead, undermining the method’s practical feasibility. The authors must:
>     a. Explore and quantify the performance of at least one low-cost proxy for model uncertainty (e.g., Monte Carlo Dropout, Evidential Deep Learning) to demonstrate a pathway toward a single-model, scalable solution.
>     b. Include a transparent cost–benefit analysis (e.g., a table comparing F1 score vs. inference latency / memory overhead) to clearly quantify the computational trade-off introduced by the ensemble.
>
> We implemented an MC dropout-based approach and added resource usage analysis in Appendix F. Switching from ensemble to dropout reduces training time but slightly degrades F1 score. The results suggest that, even with some F1 loss, the method still outperforms the no-weighting baseline, and the trade-off may be acceptable in practice.
>
> 2. Address Generalization Fragility (Vision Data Failure). The method’s failure on the BenthicNet-E vision dataset requires a deeper, mechanism-focused explanation rather than surface-level reporting. Specifically:
>     a. Discuss the interaction between the highly weighted semantic loss, CNN gradient dynamics, and augmentation strategies.
>     b. Identify the failure modes responsible for the degradation in visual domains.
>     c. Suggest explicit boundary conditions under which the proposed method is expected to perform reliably (e.g., tasks involving semantic or discrete hierarchies).
>
> Our hypothesis is that the approach provides the greatest benefits in challenging contexts where rare nodes are a problem. We have revised our text to more clearly reflect this point. The advantages diminish if the rare nodes are not a significant problem in the first place. To provide deeper insight, we added two new analyses:
>
> First, in Appendix I, we performed a node-based analysis for both untrained and fully trained encoders. Our approach consistently improves performance on deeper nodes, even though, for fully trained encoders, the gains are smaller and are offset by minor losses at higher hierarchy levels, resulting in similar overall F1 scores. Thus, the intended effect of our method remains present.
>
> Second, Appendix J presents a study that restricts the percentage of available training data. We find that our approach provides the most significant gains in low data domains. As available data grows, the relative benefit diminishes, as we might expect.
>
> 3. Address Optimization Stability ($\tilde{w}0$ Sensitivity). The hyperparameter currently serves as a manual control over the precision-recall trade-off, which raises concerns about optimization stability. The authors must:
> a. Provide a rigorous discussion of $\tilde{w}0$ as a sensitivity factor and its impact on convergence.
> b. Explore methods to dynamically learn or adapt $\tilde{w}0$ during training, or
> c. Present empirical evidence that the objective function remains stable across a broad, practically relevant range of values.
>
> To assess sensitivity and robustness, we tested a range of values on the Expr (FUN) dataset, in Appendix D. As $\tilde{w}_0$ increases, precision rises logarithmically while recall declines exponentially, explaining the observed benefit of small values. We also attempted dynamic scheduling for weights, but did not see notable improvements (Appendix D). Combining imbalance weighting and LPROS did, however, increase precision and F1.
>
> Strengthening Revisions (Recommended):
>
> 1. Wider Domain Testing. Evaluate the proposed method on an additional, distinct data modality (e.g., text classification hierarchy) to provide stronger evidence of generalizability beyond the current genomic datasets.
>
> We have added experiments on Enron (language) and Diatom (unicellular algae image) datasets (Appendix G). Our methods provide a clear benefit in Enron, and improved recall in Diatom, supporting cross-domain generalizability.
>
> 2. Detailed Ablation Analysis. Include a supplementary ablation study focusing on the performance of common (frequent) nodes to confirm that the significant recall gains observed on rare nodes do not cause catastrophic underfitting or reduced predictive quality on other parts of the hierarchy.
>
> Our node-level analysis (see above) also attempts to address this point. Gains on rare nodes do not result in significant degradation for common nodes.. However, in the trained encoder, there may be small degradations for common nodes offset by small gains for rare nodes.
>
> 3. Reproducibility Documentation. Explicitly state recommended initialization or starting ranges for key hyperparameters ($U_0$, $k$, $\tilde{w}_0$) to lower the barrier for independent reproduction and reliability validation.
>
> We have updated our hyperparameters section (Appendix A) to specify these values.

---

### Review · Reviewer_i5Ky · 2025-11-03

**Summary Of Contributions:**

The paper presents a novel training objective for rare-node detection in hierarchical multi-label classification (HML). It extends the hierarchy-consistent loss (C-HMCNN) by introducing two multiplicative gates:

1.	Node-wise imbalance weighting — using a learnable floor term ($\tilde{w}_0$) to reweight rare nodes.
2.	Uncertainty-driven focal weighting — derived from ensemble-based uncertainty (bBMA, GMU, epistemic), controlled by an exponent k.

The combined objective encourages confident predictions for uncertain rare nodes while maintaining hierarchical consistency. Experiments on gene-function (FUN/GO) and benthic imagery datasets show improved recall and F1 for rare nodes compared to hierarchy-aware baselines (HROS-PD, LPROS, etc.).

Strength:

1. Well-motivated rare-node focus: The problem of hierarchical imbalance is important yet underexplored, and the modular weighting design directly addresses this.
2. Conceptual clarity: The two-gate formulation ($\tilde{W}(Y,f)$ and $U(\Theta(X))^k$) makes the method interpretable and easily extensible.
3. Strong recall improvements: Particularly in the FUN dataset, the method achieves up to $5×$ recall improvement over the base hierarchy-consistent loss.
4. Comprehensive uncertainty treatment: Evaluates multiple ensemble-based uncertainty types (bBMA, GMU, epistemic) and shows consistent recall/F1 gains with larger ensemble sizes.
5. Reproducibility: Equations, algorithms, and ablation analyses are clearly presented; easy to implement atop existing HML models.

Weakness:

1. Limited generalization – Most results are on gene-product datasets; on the benthic imagery dataset, gains shrink or vanish when strong pretrained encoders are used (§5.4). This limits generality across visual hierarchies.
2. Ensemble dependency – The uncertainty term requires ensembles of size 10, which raises computational cost during both training and inference. The paper lacks compute/performance trade-off analysis.
3. Precision–recall trade-off – While recall improves, precision often drops in several setups. The authors do not quantify the severity or propose mitigations (e.g., dynamic reweighting schedules).
4. Incomplete sensitivity analysis – The impact of hyperparameters ($\tilde{w}_0, k, U_0$) and differences between ensemble types (bBMA vs. GMU vs. epistemic) are only partially explored.
5. Comparative positioning – The discussion of why node-wise reweighting outperforms observation-wise schemes (e.g., HROS-PD) remains superficial.
6. Scalability – No experiments on large DAGs or text/image hierarchies beyond small-scale datasets; unclear how the method behaves under noise or multi-parent nodes.

**Audience:**

Yes

**Audience Explanation:**

The paper targets readers in hierarchical learning, uncertainty modeling, and imbalanced classification, aligning with the TMLR audience.

**Broader Impact Concerns:**

Nil

**Claims And Evidence:**

Yes

**Claims Explanation:**

Claims about recall improvement and uncertainty-driven adaptation are supported by consistent quantitative evidence.

**Requested Changes:**

Please go through the Weakness section to find the detail suggestions and changes. In brief, the suggested changes are as follows:

1. Compute overhead: What are the relative training/inference costs for ensemble size $10$ vs. single models? Can MC-Dropout or snapshot ensembles offer similar uncertainty signals?
2. Failure regimes: Under what imbalance ratios or hierarchy depths does the precision degrade? Can a dynamic $\tilde{w}_0$ scheduling alleviate this?
3. Ablation depth: Please provide sensitivity plots for $\tilde{w}_0$, $k$, and $U_0$ across both datasets to show robustness.
4. Broader benchmarks: Can you validate the method on larger-scale HML benchmarks (e.g., iNaturalist, OpenImages-Hierarchy) to confirm the claimed generality?
5. Calibration and thresholding: Have you evaluated calibration metrics (ECE, Brier) to ensure that improved recall is not simply due to lower thresholds on ensemble-averaged outputs?

---

> ### Author Response · Authors · 2025-11-25
> **Rebuttal for Reviewer i5Ky**
>
> Thank you for your insightful comments. We address each point below.
>
> 1. Compute overhead (Ensemble dependency). What are the relative training/inference costs for ensemble size vs. single models? Can MC-Dropout or snapshot ensembles offer similar uncertainty signals?
>
> We have added an experiment in Appendix F, addressing compute and performance trade-off between ensembles and MC-dropout.
>
> 2. Failure regimes (Precision–recall trade-off). Under what imbalance ratios or hierarchy depths does the precision degrade? Can a dynamic $\tilde{w}_0$ scheduling alleviate this?
>
> In Appendix D, we developed a dynamic scheduler shifting imbalance weighting to a neutral form during training. However, this did not perform strongly. We also combined weighting with LPROS, which produced stronger results.
>
> We have also clarified our claim in that the issue isn't so much the encoder, but the difficulty of the task for the model. Our approach yields strong benefits in challenging contexts, but diminishes if rare nodes aren't a significant challenge in the first place. In Appendix I, node performance is examined individually. Even in the trained encoder, performance is slightly improved for rare nodes, while high-level nodes show a slight offset, balancing out overall. We further added a data ablation study in Appendix J. In low data regimes, our method holds significant advantages over baseline. This advantage diminishes with more data, as expected.
>
> 3. Ablation depth (Incomplete sensitivity analysis). Please provide sensitivity plots for $\tilde{w}_0$, $k$, $U_0$ and across both datasets to show robustness.
>
> We have added sensitivity plots in Appendix H. We observe that recall decays exponentially and precision increases slower with increasing $\tilde{w}_0$, allowing small values to achieve strong recall with minimal precision trade-off. For $u_0$ and $k$, the impact is more stable and modest: above $0.2$, $u_0$ reduces recall for values while increasing precision, and a larger $k$ decreases precision but increases recall.
>
> 4. Broader benchmark (Limited generalization). Can you validate the method on larger-scale HML benchmarks (e.g., iNaturalist, OpenImages-Hierarchy) to confirm the claimed generality?
>
> We currently lack the compute to quickly process datasets of this scale (hundreds of thousands to millions of images). Additionally, familiarizing ourselves and applying our approach would require substantial effort, possibly shifting the paper’s focus away from smaller data regimes common to specialized research groups, where imbalance is most problematic.
>
> For generalization, we test on two more datasets in Appendix G. Enron is a language dataset, and Diatom is an image dataset for unicellular algae. We observe clear benefits in Enron and more muted, but still statistically significant, recall improvement in Diatom. This pattern is consistent with our other datasets.
>
> 5. Scalability. No experiments on large DAGs or text/image hierarchies beyond small-scale datasets; unclear how the method behaves under noise or multi-parent nodes.
>
> The GO datasets we use (see Appendix C) are DAGs, featuring multi-parent hierarchies and significant complexity (~4000 nodes). Our method addresses imbalance in these multi-parent settings naturally, unlike HROS-PD. Our goal is to help research groups with limited specialized data and annotations, where imbalance is a key challenge. With more data, imbalance becomes less problematic.
>
> 6. Calibration and thresholding. Have you evaluated calibration metrics (ECE, Brier) to ensure that improved recall is not simply due to lower thresholds on ensemble-averaged outputs?
>
> An interesting point! While perhaps true, we do not consider it problematic. Arbitrary thresholding is a historical issue in multi-label literature, even though thresholding is necessary for predictions. If optimal thresholds per node were available, recall and overall performance might indeed improve, possibly matching our approach. However, determining ideal thresholds is nontrivial.
>
> Calibration might approach similar results, perhaps via a calibrated loss. Extending calibration from multi-class to multi-label likely requires treating each node as its own classification problem with its own temperature parameter, but integrating hierarchical integrity is less clear.
>
> 7. Comparative positioning. The discussion of why node-wise reweighting outperforms observation-wise schemes (e.g., HROS-PD) remains superficial.
>
> A key reason node-wise reweighting can outperform observation-based is its flexibility to focus on annotation parts, not whole observations, which in HML are always multi-label. Emphasizing an observation with rare nodes also accentuates common parent nodes, which is suboptimal. Detaching these connections allows the model to focus on rare nodes exclusively.

---

### Review · Reviewer_f5e4 · 2025-11-17

**Summary Of Contributions:**

**Contributions**

The paper proposes a composite loss formulation that integrates node-wise imbalance weighting and uncertainty-based focal weighting to improve detection of rare nodes in hierarchical multi-label learning. The authors make two main claims: (a) the proposed loss improves detection of rare nodes across HML datasets, and (b) it also benefits vision CNNs when the encoder is not specialized for the downstream task.


**Strengths**
  - Novel methodological idea: Authors propose weighting rare nodes directly instead of resampling observations, which differs from prior HML imbalance methods.
  - Core claim (improving detection of rare nodes) is well supported.

**Weaknesses**
  - _The problem is not motivated._ The paper never clearly argues why rare node detection is practically important, nor why existing approaches are inadequate.
  - _Method and background lack rigor._ Notation requires the reader to infer missing definitions (e.g., domains, tensor shapes).
  - _Hard to follow._ The prose often reads like a stream-of-consciousness log of what was tried rather than a structured, self-contained presentation of the contributions.
  - _Second claim is not well supported_ (see below).
  - _No discussion of limitations._

**Audience:**

No

**Audience Explanation:**

The paper does not convincingly motivate why rare-node detection in HML matters in practice, nor does it show that existing approaches fail in applications. Without a concrete use case or previously demonstrated practical relevance, it is unclear to me why the TMLR audience should be interested in the findings.

**Broader Impact Concerns:**

None.

**Claims And Evidence:**

No

**Claims Explanation:**

The second claim—that the joint loss benefits vision CNNs when encoders are not fully adapted to the task—is not convincingly supported by the experiments. The vision evaluation uses only a single, small dataset, and the “weak encoder” condition is created by literally linearly mixing random and pretrained weights, which seems like an artificial degradation that does not correspond to any realistic training process.

**Requested Changes:**

1. [necessary] _Motivate the problem._ The introduction only mentions that class imbalance in HML is an under-explored area. Is there a real use case where rare node detection in HML affects downstream tasks? Is there evidence that current methods are inadequate?
2. [necessary] _Clarify notation and problem statement._
    - The variables in Sections 2–3 lack basic definitions such as their domains (possible values) and shapes (scalar/vector/matrix, with dimensions).
    - Define the data and explicitly state the prediction problem.
3. [necessary] _Provide evidence (empirical/analytical or citation) for the following claims_:
   - "Observation-wise imbalance weighting risks over-emphasizing common components of an annotation […] therefore we conduct weighting on a node-wise basis."
   - "Regardless of formulations for negative weightings, we were unable to achieve strong performance."
4. [necessary] _Expand CNN vision experiments to validate second main claim_
    - either include realistic partially specialized encoders (e.g. fine-tuning a pretrained model with only the last block unfrozen) OR report basic sanity metrics across different training factors.
    - add more CNN backbones (smaller and/or larger ResNet).
    - test at least one more hierarchical vision datasets or clearly state that conclusions may not generalize beyond BenthicNet-E.
5. [necessary] _Discuss limitations and computational complexity._
    - What are the limitations of the method? Does the loss increase training cost (especially the focal weighting component)?
    - How do the limitations and complexity compare to resampling approaches?

---

> ### Author Response · Authors · 2025-11-25
> **Rebuttal for Reviewer f5e4**
>
> Thank you for your advice and time in providing valuable feedback for our work.
>
> 1. Motivate the problem. The introduction only mentions that class imbalance in HML is an under-explored area. Is there a real use case where rare node detection in HML affects downstream tasks?
>
> To better capture our motivations, we have added the following to the introduction:
>
> Rare nodes often represent fine-grained distinctions crucial for research. For example, in seafloor classification, one of our areas of interest, detecting rare species or habitats, can indicate important environmental changes. This valuable information may then be used to guide government policies concerning maritime economies. Similarly, in medical domains, rare gene products identified through hierarchical classification can help diagnose disease. Thus, handling imbalance and ensuring that rare nodes are meaningfully incorporated into HML models is essential for producing insights that are both scientifically relevant and actionable.
>
> "Is there evidence that current methods are inadequate?"
>
> The FUN and GO datasets are well-known benchmarks, yet remain unsolved even with modern HML and oversampling methods. As shown in surrounding literature and our results, most nodes remain unpredicted, with low F1 scores. Our work addresses this persistent under-detection in HML classification.
>
> 2. Clarify notation and problem statement. The variables in Sections 2–3 lack basic definitions such as their domains (possible values) and shapes (scalar/vector/matrix, with dimensions). Define the data and explicitly state the prediction problem.
>
> We added dimensional information to the definitions, clarifying the data and problem.
>
> 3. Provide evidence (empirical/analytical or citation) for the following claims:
>
> "Observation-wise imbalance weighting risks over-emphasizing common components of an annotation […] therefore we conduct weighting on a node-wise basis."
>
> We now state at the end of our introduction:
>
> By emphasizing the entire observation, on the basis that it contains a rare component in its annotation, additional emphasis on the already common components becomes nearly inevitable. Therefore, an observation-based approach is suboptimal in that it must also add to a problem it is simultaneously attempting to solve. Instead, we propose looking at the problem through a node-based lens, where the emphasis on nodes is independent of the distribution of observations.
>
> "Regardless of formulations for negative weightings, we were unable to achieve strong performance."
>
> This was one of many preliminary experiments, reported to inform the research community and reduce redundant efforts. Performance degradation with negative weights likely arises from interactions between weight sets in the loss. While other formulations might be possible, our attempts did not yield successful results and are not part of our main approach.
>
> 4. Expand CNN vision experiments to validate second main claim. Either include realistic partially specialized encoders (e.g. fine-tuning a pretrained model with only the last block unfrozen) OR report basic sanity metrics across different training factors. Add more CNN backbones (smaller and/or larger ResNet). Test at least one more hierarchical vision datasets or clearly state that conclusions may not generalize beyond BenthicNet-E.
>
> We have clarified with revisions in our paper that the goal is to demonstrate benefits in challenging environments. Linear mixing simulates more difficult tasks by adding noise to the encoder.
>
> Following your suggestions, we conducted additional experiments in Appendix J, where we increased the difficulty by restricting training data from 1% to 30% of the total. The significant initial F1 advantage from our method diminishes as more data becomes available, as expected. This pattern holds for both ResNet-18 and ResNet-50 backbones.
>
> We have also added results for two additional datasets: “Enron” (text-based email classification) and “Diatoms” (unicellular algae image classification). Both show similar effects to our main datasets, supporting generalization claims.
>
> 5. Discuss limitations and computational complexity. What are the limitations of the method? Does the loss increase training cost? How do the limitations and complexity compare to resampling approaches?
>
> We added Appendix F (compute costs, performance comparison: ensemble vs. MC dropout) and Appendix H (parameter sensitivity). These results clarify how parameters affect performance limitations in the precision-recall trade-off and assist researchers in parameter selection.
>
> Resampling methods are cheap, being a one-time preprocessing cost. However, direct comparison to loss-based uncertainty approaches is difficult, as costs depend on different factors: dataset size for resampling, ensemble size and uncertainty calculation for loss-based methods. Variations in specific approaches can further affect cost, and it is difficult to make general claims.

---

### Decision · Action_Editor_fWwX · 2026-01-25

**Recommendation:** Accept with minor revision

**Additional Comments:**

The paper just needs to be clearer. I can check that myself without going through a new review process.

**Audience:**

Yes

**Audience Explanation:**

The paper works on the are nodes in hierarchical multi-label classification. It is an underexplored problem, and the paper makes a valid contribution.

**Claims And Evidence:**

Yes

**Claims Explanation:**

The reviewers of this paper acknowledge that the authors make a valuable contribution to the problem of detecting rare nodes in hierarchical multi-label classification. The authors propose a two-component weighted loss framework that combines node-wise imbalance weighting with focal weighting based on modern ensemble uncertainty quantification.  The experimental evaluation is comprehensive, spanning 16 gene product benchmark datasets across two hierarchical schemes (FUN and GO) and image classification tasks on marine imagery. The results are compelling: recall improvements of up to 5x over existing methods, with statistically significant F1 gains across datasets. The paper also adds a thought ablation study.

One of the reviewers says the paper is hard to follow and would benefit from significant rewriting. I suggest the authors consult a language expert or an LLM to improve their writing. If using the latter, please do not simply copy and paste; ensure the paper remains in the authors' voice.

---

> ### Author Response · Authors · 2026-01-30
> **Response for AE**
>
> Thank you, Dr. Perez-Cruz, for the time and effort you put into reviewing our paper. We have since uploaded the deanonymized camera-ready version of this paper, along with a slightly earlier version highlighting (in blue) the main modifications we made in an effort to clarify our work as per the requested minor revisions. The latter has been submitted under "Supplementary Materials".
>
> Although we edited the manuscript throughout, we made the most dramatic changes in the areas we believe Reviewer f5e4 was most likely alluding to, specifically our methods section and in some of our analyses in the experiments sections. The difference between the main camera-ready version and the one highlighting changes includes a few additional minor edits to wording, clarity, and accuracy of our statements, and a change in formatting for the $F_{1}$ score.
>
> Upon deanonymizing the work, our paper has increased by about half a page in length, due to listing the authors and the acknowledgement section, thus exceeding the 12-page limit for the TMLR quick-turnaround review process we enjoyed. Please let us know if we should shorten it.
>
> We are sincerely grateful for all the effort both you and the reviewers have put into helping shape our work!